# INVESTIGATING THE BENEFITS OF PROJECTION HEAD FOR REPRESENTATION LEARNING

**Yihao Xue, Eric Gan, Jiayi Ni, Siddharth Joshi, Baharan Mirzasoleiman**
Department of Computer Science,
University of California, Los Angeles
`yihaoxue@g.ucla.edu, egan8@g.ucla.edu, nijiayi1119626@g.ucla.edu,`
`sjoshi804@cs.ucla.edu, baharan@cs.ucla.edu`

## ABSTRACT

An effective technique for obtaining high-quality representations is adding a projection head on top of the encoder during training, then discarding it and using the pre-projection representations. Despite its proven practical effectiveness, the reason behind the success of this technique is poorly understood. The pre-projection representations are not directly optimized by the loss function, raising the question: what makes them better? In this work, we provide a rigorous theoretical answer to this question. We start by examining linear models trained with self-supervised contrastive loss. We reveal that the implicit bias of training algorithms leads to layer-wise progressive feature weighting, where features become increasingly unequal as we go deeper into the layers. Consequently, lower layers tend to have more normalized and less specialized representations. We theoretically characterize scenarios where such representations are more beneficial, highlighting the intricate interplay between data augmentation and input features. Additionally, we demonstrate that introducing non-linearity into the network allows lower layers to learn features that are completely absent in higher layers. Finally, we show how this mechanism improves the robustness in supervised contrastive learning and supervised learning. We empirically validate our results through various experiments on CIFAR-10/100, UrbanCars and shifted versions of ImageNet. We also introduce a potential alternative to projection head, which offers a more interpretable and controllable design.

## 1 INTRODUCTION

Representation learning has been the subject of extensive study in the last decade (Chen et al., 2020; Zbontar et al., 2021; Khosla et al., 2020; Ermolov et al., 2021). Despite the great progress, learning representations that generalize well across various domains has remained challenging. Among the existing techniques, contrastive self-supervised learning has gained a lot of attention, due to its ability to learn robust representations that generalize better across various domains. Contrastive learning (CL) learns representations by maximizing the agreement between different augmented views of the same examples and minimizing that of different examples (Chen et al., 2020). Pre-training with CL is often essential before transfering models to new or shifted domains (Hendrycks et al., 2019).

A key factor that enables contrastive learning to learn robust representations is *projection head*, a shallow MLP that is used during pretraining and is discarded afterwards (Chen et al., 2020), which has become a standard for learning high-quality representations (He et al., 2020; Yeh et al., 2022; Bardes et al., 2022; Chuang et al., 2020; Grill et al., 2020; Garrido et al., 2022; Zbontar et al., 2021; Ermolov et al., 2021; Joshi & Mirzasoleiman, 2023). It is particularly advantageous when there is a misalignment between training and downstream objectives (Bordes et al., 2023). The benefit of projection head extends beyond self-supervised learning to other representation learning methods, including supervised contrastive learning (Khosla et al., 2020). However, the mechanism by which the projection head improves the generalizability and robustness of representations is poorly understood.

Theoretically analyzing the effect of projection head for CL is challenging, as one needs to understand feature learning both before and after the projection head and compare them. A few recent studies (Tian et al., 2021; Wang et al., 2021; Wen & Li, 2022) have focused on non-contrastive self-supervised methods and demonstrated that the projection head can mitigate the dimensional collapse problem, where all inputs are mapped to the same representation. However, such results

do not extend to contrastive methods due to its inherently different learning mechanism. Besides, they do not provide an understanding of the projection head's role in enhancing robustness under misalignment between pretraining and downstream objectives.

In this work, we analyze the effect of projection head on the quality and robustness of representations learned by CL, and extend our results to supervised CL (SCL) and supervised learning (SL). First, we theoretically show that linear models progressively assign weights to features as they operate through the layers. Thus, in deeper layers the features are represented more unequally and the representations are more specialized toward the pretraining objective. Moreover, non-linear activations exacerbate this effect allowing lower layers to learn features that are entirely absent in the outputs of the projection head. We demonstrate that projection head provably improves the robustness and generalizability of the representations, when data augmentation harms useful features of the pretraining data, or when features relevant to the downstream task are too weak or too strong in the pretraining data. Finally, we extend our results beyond CL to SCL and SL. In this setting, we reveal that lower layers can learn subclass-level features that are not represented in the final layer, demonstrating how representations before the final representation layer can significantly reduce class/neural collapse, a problem previously observed in the final representations.

We conduct extensive experiments to confirm our theoretical analysis. First, we construct a semi-synthetic dataset by adding MNIST (LeCun, 1998) digits to CIFAR10 (Krizhevsky et al., 2009) images, and confirm that using projection head yields superior representations when data augmentation harms the downstream-relevant features, or when these features are either very strong or very weak during pretraining. Then, we study supervised learning and demonstrate that using projection head results in better course-to-fine transferability on CIFAR100, superior performance of few-shot adaption to distribution shift on UrbanCars (Li et al., 2023), and better robustness against natural distribution shifts in ImageNet (Xiao et al., 2020; Hendrycks et al., 2021a;b). We also demonstrate how a fixed reweighting head can achieve performance comparable to the projection head, providing further evidence for our theoretical conclusions and potentially inspiring future design.

## 2 ADDITIONAL RELATED WORK

**Projection head.** While projection head is widely used, the underlying reasons for its benefit have not been fully understood. The empirical study by Bordes et al. (2023) suggests that the benefit of the projection head is influenced by optimization, data, and downstream task, and is especially significant when there is a misalignment between the training and downstream tasks. On the theoretical side, Jing et al. (2021) suggests that projection head can alleviate dimensional collapse, but did not provide insights into the significant role of the projection head in cases of misalignment between pretraining and downstream tasks. Recently, Gui et al. (2023) analyzed the training of a linear projection head using CL. However, their analysis is performed in the case where representations are fixed, and the results only show how training a linear head on these representations can lead to worse post-projection representations. This does not align with practical scenarios where encoder and projection are trained simultaneously and does not reveal why adding a projection head is needed in the first place.

**Generalizability and transferability of representations.** A longstanding goal in machine learning is to acquire representations that generalize and transfer across various tasks. Two key challenges emerge in this pursuit. The first arises from shifts in labels between training and downstream tasks. Notably, the issue of class or neural collapse (Papyan et al., 2020; Han et al., 2021; Zhu et al., 2021; Zhou et al., 2022b;a; Lu & Steinerberger, 2022; Fang et al., 2021; Hui et al., 2022; Chen et al., 2022; Graf et al., 2021; Xue et al., 2023), affecting both SCL and SL, where representations within the same class become indistinguishable at a subclass level, rendering them unfit for fine-grained labeling. We theoretically demonstrate that such issue can be alleviated by taking the pre-projection head representations. The second challenge arises from shifts in input distributions, where neural networks rely heavily on patterns specific to the training distribution that do not generalize (Zhu et al., 2016; Geirhos et al., 2018; Ilyas et al., 2019; Barbu et al., 2019; Recht et al., 2019; Sagawa et al., 2019; 2020; Xiao et al., 2020; Taori et al., 2020; Koh et al., 2021; Shankar et al., 2021). We explore the benefits of projection head in the above two scenarios, but noting that our primary focus lies in elucidating the broader concepts at play rather than addressing each specific problem.

## 3 EXPLORING THE ROLE OF PROJECTION HEAD IN SELF-SUPERVISED CL

In this section we consider self-supervised contrastive learning, a scenario where the concept of the projection head is frequently employed and has a crucial role in achieving optimal performance.

**Pretraining data distribution.** We define the following data distribution and data augmentation used for pretraining with contrastive learning. We choose this setting for clarity in presenting our results, but we note that our results can be generalized to more complex cases.

**Definition 3.1** (Input distribution of pretraining data). *The input data used for pretraining follows distribution $\mathcal{D}$, where, for an input $\boldsymbol{x} \in \mathbb{R}^d$, its $i$-th element is randomly drawn from $\{-\phi_i, \phi_i\}$.*

**Definition 3.2** (Data augmentation). *Given an input $\boldsymbol{x}$, its augmentation follows the distribution $\mathcal{A}(\boldsymbol{x})$. The augmentation operates on an input $\boldsymbol{x}$ in two steps. First, for each $i$, with probability $\alpha_i$, it alters $i$-th element in $\boldsymbol{x}$ by randomizing its sign, resulting in a modified input denoted by $\boldsymbol{x}'$. Then, a random noise $\boldsymbol{\xi}$ satisfying $\mathbb{E}[\boldsymbol{\xi}] = \boldsymbol{0}$ and $\mathbb{E}[\boldsymbol{\xi}\boldsymbol{\xi}^\top] = \sigma^2 \mathbf{I}$ is added to $\boldsymbol{x}'$, yielding augmentation $\boldsymbol{x}' + \boldsymbol{\xi}$.*
Each coordinate of the input is an independent feature. At a high level, $\phi_i$ represents the magnitude of a feature in the input data, while $\alpha_i$ quantifies the level of disruption introduced by the augmentation. If $\alpha_i = 1$, it means that this feature is completely 'destroyed' during augmentation, such that there is no correlation between a positive pair (two augmented versions of the same input) at this coordinate. This data augmentation is designed to mimic practical scenarios where data augmentation techniques intentionally modify different features to varying degrees. For example, some features, like color and texture, may be altered more significantly than core features of the main object. Additionally, the noise $\boldsymbol{\xi}$ with variance $\sigma^2$ at each coordinate is included because data augmentation used in practice unintentionally introduces certain level of noise.

**Model.** We consider 2-layer linear and non-linear models for the sake of clarity, as it suffices to demonstrate our main findings, although our results generalize to multiple layers. Given an input $\boldsymbol{x} \in \mathbb{R}^d$, the output after the $l$-th layer is denoted as $f_l(\boldsymbol{x})$, with $f_2(\boldsymbol{x}) = h(f_1(\boldsymbol{x}))$. We will consider linear and non-linear functions for $f_1(\cdot)$ and $h(\cdot)$ in our analysis. The second layer $h(\cdot)$ serves as the projection head and the first layer $f_1(\cdot)$ serves as the encoder.

**Contrastive loss.** We consider the following spectral loss, which has been widely used in previous theoretical and empirical studies (HaoChen et al., 2021; Xue et al., 2022; Saunshi et al., 2022; HaoChen & Ma, 2022; Garrido et al., 2022; Xue et al., 2023). Given a model representing function $f(\cdot)$, the loss is
$$\mathcal{L}_{CL}(f) = -2\mathbb{E}_{\boldsymbol{x}\sim\mathcal{D},\boldsymbol{x}_1^+\sim\mathcal{A}(\boldsymbol{x}),\boldsymbol{x}_2^+\sim\mathcal{A}(\boldsymbol{x})}[f(\boldsymbol{x}_1^+)^\top f(\boldsymbol{x}_2^+)] + \mathbb{E}_{\boldsymbol{x}_1\sim\mathcal{D},\boldsymbol{x}_2\sim\mathcal{D}}[(f(\mathcal{A}(\boldsymbol{x}_1))^\top f(\mathcal{A}(\boldsymbol{x}_2)))^2].$$

After training a model to minimize this loss, we utilize it in *downstream tasks* where we feed different inputs, which may or may not follow the same distribution as the pretraining data, into the model. We can leverage the *representations* provided by the model for various purposes, most commonly to train a linear model on these representations and predict labels for the downstream data. When we use the two-layer model, i.e., when $\mathcal{L}_{CL}(f_2)$ is minimized, we have the option to choose either the post-projection representations generated by $f_2(\cdot)$ or the pre-projection representations from $f_1(\cdot)$ for the downstream task.

**Clarification on what we compare.** To gain a comprehensive understanding, it's necessary to compare these three cases: (1) *pre-projection*, where we minimize $\mathcal{L}_{CL}(f_2)$ first and discard $h(\cdot)$, using only $f_1(\cdot)$ for the downstream task; (2) *post-projection*, where we minimize $\mathcal{L}_{CL}(f_2)$ and use $f_2(\cdot)$ for the downstream task; (3) *no-projection*, where we minimize $\mathcal{L}_{CL}(f_1)$ and using $f_1(\cdot)$ for the downstream task. The goal is to determine when *pre-projection* outperforms both of the other two and understand the reasons behind it. However, in practical deep neural networks, there is typically no significant difference between *no-projection* and *post-projection* as both scenarios use the network's final output for the downstream task and the difference is mainly an additional layer in *post-projection*. Given that the networks are sufficiently large and expressive, this one-layer difference does not significantly change the representations achieved at the output. Therefore, in theory, we consider settings where $f_1(\cdot)$ and $f_2(\cdot)$ have the same expressiveness and yield equivalent representations when used to minimize the loss, and then solely comparing *pre-projection* and *post-projection* is sufficient.

### 3.1 LAYER-WISE PROGRESSIVE FEATURE WEIGHTING IN LINEAR MODELS

**Linear network.** We consider a linear model in which $f_2(\boldsymbol{x}) = \boldsymbol{W}_2 f_1(\boldsymbol{x}) = \boldsymbol{W}_2\boldsymbol{W}_1\boldsymbol{x}$, with $\boldsymbol{W}_1 \in \mathbb{R}^{p\times d}$ and $\boldsymbol{W}_2 \in \mathbb{R}^{p\times p}$ representing the weights of the first and second layers, respectively. Both the hidden and output dimensions are $p$, to ensure consistent dimensionality between pre-projection and post-projection representations for a fair comparison.

#### 3.1.1 STRUCTURE OF LAYER WEIGHTS

We begin by examining the weights at different layers within a model. Our investigations reveals the relationship between these layer weights, which generally hold regardless of the data distribution.

Firstly, we consider the weights of the minimum norm minimizer of the CL loss. This is pertinent because gradient-based algorithms are shown to prefer minimizers with small norms (Neyshabur et al., 2014; Gunasekar et al., 2017). Furthermore, many theoretical studies on CL (Ji et al., 2021; Liu et al., 2021; Nakada et al., 2023) have considered regularization in the form of $\|\boldsymbol{W}^\top \boldsymbol{W}\|_F$, which promotes a small norm, and Xue et al. (2023) have shown that the minimum norm provides an explanation for many intriguing phenomena in CL.

**Theorem 3.3** (Weights of the minimum norm minimizer). *The global minimizer of the CL loss $\mathcal{L}_{CL}$ with the smallest norm, defined as $\|\boldsymbol{W}_1^\top \boldsymbol{W}_1\|_F^2 + \|\boldsymbol{W}_2^\top \boldsymbol{W}_2\|_F^2$, satisfies $\boldsymbol{W}_1 \boldsymbol{W}_1^\top = \boldsymbol{W}_2^\top \boldsymbol{W}_2$.*

In addition, we establish that a similar conclusion holds for models trained using gradient flow, which is a continuous version of gradient descent widely adopted in theoretical analysis. In gradient flow, at any time $t$, the weight updates are given by $\frac{d}{dt}\boldsymbol{W}_i^{(t)} = -\frac{\partial}{\partial \boldsymbol{W}_i^{(t)}}\mathcal{L}_{CL}(\boldsymbol{W}^{(t)})$, where $i = 1, 2$.

**Theorem 3.4** (Weights of the model trained with gradient flow, proved in (Arora et al., 2018)). *Suppose the initialization satisfies $\boldsymbol{W}_1^{(0)}\boldsymbol{W}_1^{(0)\top} = \boldsymbol{W}_2^{(0)\top}\boldsymbol{W}_2^{(0)}$. Using gradient flow, at any time $t$, we have $\boldsymbol{W}_1^{(t)}\boldsymbol{W}_1^{(t)\top} = \boldsymbol{W}_2^{(t)\top}\boldsymbol{W}_2^{(t)}$.*

### 3.1.2 LAYER-WISE PROGRESSIVE FEATURE WEIGHTING

What insights can we gain from Theorems 3.3 and 3.4? Given that $\boldsymbol{W}_1\boldsymbol{W}_1^\top = \boldsymbol{W}_2^\top\boldsymbol{W}_2$, both layers have the same singular values, and the left singular vectors of $\boldsymbol{W}_1$ match the right singular vectors of $\boldsymbol{W}_2$. Consequently, the singular values of the joint weight matrix $\boldsymbol{W}_2\boldsymbol{W}_1$ are the squares of those in the first layer, $\boldsymbol{W}_1$. As a result, the differences in weights assigned to the features are smaller when the input is passed through the first layer than when the input is passed through the whole network. To illustrate this concept, we analyze the model trained on the data distribution given by Definition 3.1 and analyze the resulting representations.

The following analysis holds for both models obtained from either the minimum norm minimizer of the loss (as in Theorem 3.3) or the model trained using gradient flow under the assumption that the model converges to a global minimum (as in Theorem 3.4), as they are equivalent.

In our input data, we refer to the $d$ independent coordinates as input features. Our interest lies in understanding the weight assigned to each feature in the pre- and post-projection representations. To achieve this, we examine the quantities $\|f_l(\boldsymbol{e}_i)\|$, $i = 1, \ldots, d$, and $l = 1, 2$, with $\boldsymbol{e}_i$ denoting $i$-th standard basis. These quantities represent the scale of the representation of a unit feature at each coordinate. The following theorem (see proof in Appendix A.2) shows the weights of the features in each layer:

**Theorem 3.5.** *Define $\beta_i := \frac{(1-\alpha_i)^2 \phi_i^2}{\phi_i^2 + \sigma^2}$ and $\gamma_i := \sqrt{\frac{(1-\alpha_i)\phi_i}{\phi_i^2 + \sigma^2}}$. Let $\Pi := (j_1, j_2, \ldots, j_d)$ be a permutation of indices $\{1, 2, \ldots, d\}$ such that $\beta_{j_1} \geq \cdots \geq \beta_{j_d}$. Then after pretraining,*

$$\|f_l(\boldsymbol{e}_i)\| = \gamma_i^l \quad \text{if } i \in \{j_1, \ldots, j_{\min\{d,p\}}\}, \quad \text{else } 0.$$

**What does the model do?** According to this theorem, the model follows two key steps: (1) Feature selection: The model selects the top $p$ features with the highest $\beta_i$ values, which experience a low level of disruption from augmentation ($\alpha_i$) and/or have a large feature magnitude ($\phi_i$). (2) Feature weighting: These selected features are scaled by $\gamma_i$ at each layer, with zero weight assigned to the remaining features. The rescaling serves a dual purpose: (a) *moderating* the features by assigning small weights to either overly strong or overly weak features in terms of their magnitude, as indicated by $\gamma_i \to 0$ when $\phi_i$ approaches either 0 or $+\infty$; (b) giving larger weights to features that are less disrupted by augmentation.

**The difference between $f_1(\cdot)$ and $f_2(\cdot)$.** (1) Both learn the same features but assign different weights to them. (2) $f_1(\cdot)$ treats features more equally, exhibiting a smaller gap between feature weights.

### 3.1.3 WHY AND WHEN CAN MORE NORMALIZED FEATURES BENEFIT A DOWNSTREAM TASK?

To address this question, we analyze the representations of downstream data drawn from the following distribution: each input $\boldsymbol{x} \in \mathbb{R}^d$ is a vector, and its $i$-th element is independently drawn from the set $\{-\hat{\phi}_i, \hat{\phi}_i\}$. Note that $\hat{\phi}$'s may differ from $\phi_i$'s, as in real-world scenarios, the input data in the downstream task may follow a different distribution than the pretraining inputs. Additionally, each input

$\boldsymbol{x}$ is labeled as $\text{sign}(\boldsymbol{e}_{j^*}^\top \boldsymbol{x})$, where $\boldsymbol{e}_{j^*}$ is the $j^*$-th standard basis vector. In simpler terms, the label is determined by the sign of the $j^*$-th coordinate of the input, that is the *downstream-relevant feature*.

To evaluate the informativeness of the learned representations for the downstream task, we input the data (without any data augmentation, as is typically the case in practice) into the model that has been pretrained with the CL objective. We then evaluate the quality of these representations at each layer by analyzing the sample complexity of the hard SVM trained with labels on these representations, which can be equivalently viewed as training a linear model with logistic losses using gradient descent Soudry et al. (2018). This aligns with the standard linear evaluation protocol in practice (Ye et al., 2019; Oord et al., 2018; Bachman et al., 2019; Kolesnikov et al., 2019; Chen et al., 2020).For a data distribution that is separable with a $(\gamma, \rho)$-margin (see details in Appendix A.3), it is well-known that the sample complexity only grows with $r = (\rho/\gamma)^2$, Bartlett & Shawe-Taylor (1999). Hence, we refer to $r$ as the sample complexity indicator and compare its values for pre-projection and post-projection representations. The following theorem shows the conditions under which one has a higher sample complexity indicator than the other. Note that a smaller sample complexity is preferable.

**Theorem 3.6.** *Let $r_1$ and $r_2$ be the sample complexity indicators for pre-projection and post-projection representations, respectively. Define $\Delta := \sum_{1 \leq i \leq \min\{d,p\} \text{ and } j_i \neq j^*} \hat{\phi}_{j_i}^2 (\frac{\gamma_{j_i}^2}{\gamma_{j^*}^2} - \frac{\gamma_{j_i}^4}{\gamma_{j^*}^4})$. If $\Delta < 0$ then $r_1 < r_2$, and if $\Delta > 0$ then $r_1 > r_2$.*

$\Delta$ depends on the strength of the downstream-relevant feature and the weights of features. While determining $\Delta$'s value may seem complex, in general, the key factor is whether the model assigns sufficient weight to the downstream-relevant feature. If this feature is underweighted by the model, using the first layer is beneficial. To better understand when this occurs, we provide the following interpretable examples

**Corollary 3.7.** *In each of the following examples $\Delta < 0$, i.e., the pre-projection representations are preferred. (1)* **Data augmentation disrupts the useful feature too much.** *Example: all features in both the pretraining and downstream data have a magnitude of 1, and $\alpha_{j^*}$ is the p-th smallest among all $\{\alpha\}_{i=1}^d$, indicating that the data augmentation disrupts useful feature the most among the p features that will be learned. (2)* **The downstream-relevant feature is too weak in pretraining.** *Example: all $\alpha_i$'s are equal, $p \geq 2$, $\forall i \; \phi_i \leq \sigma$, and $\phi_{j^*}$ is the p-th largest among all $\{\phi_i\}_{i=1}^d$. (3)* **Multiple features are selected by the model, with the downstream-relevant being too strong in pretraining.** *Example: all $\alpha_i$'s are equal, $p \geq 2$, $\forall i \neq j^* \; \phi_{j^*} > \max\{\phi_i, \sigma/\phi_i\}$.*

It might seem surprising to include scenario 3 above, as having a strong downstream-relevant feature in pretraining may be expected to benefit the downstream task. However, as we discussed after Theorem 3.5, the model moderates the features and assigns small weights to overly strong ones. This can potentially explain why the use of a projection head remains beneficial in cases where the pretraining task seems to be a good match for the downstream task. Furthermore, we will validate each of the above three observations in experiments in Section 5.1. We also provide a discussion on multi-layer models in Appendix B.

## 3.2 Lower Layers can Learn More Features than Higher Layers Via Non-Linearity

In the previous section on linear models, it's worth noting that both layers select the same features, albeit with different weightings. Now, let's consider a scenario where augmentation disrupts the useful feature excessively, for example, when $p_{j^*} = 1$ such that the useful feature is assigned a weight of $\gamma_{j^*} = 0$. In such cases, neither layer would learn this feature. However, in this section, we'll explore an interesting aspect of non-linear models. Specifically, we'll demonstrate that pre-projection head representations can learn features that are weighted as zero in post-projection head representations.

For clarity, we will present our result in the simplest case, although it holds in broader scenarios. For the pretraining data, we let $d \geq 2$, and assume $\phi_1 = \phi_2 = 1, \sigma = 0$. Additionally, we set $p_2 = 1$ and $p_1 = 0$, *meaning that the augmentation completely 'destroys' feature 2 while fully preserving feature 1.* Consequently, during CL, the pretraining objective discourages the learning of feature 2.

**Non-Linear diagonal network.** We consider a diagonal non-linear network with

$$f_1(\boldsymbol{x}) = \sigma(\boldsymbol{w}_1 \odot \boldsymbol{x}, \boldsymbol{b}_1), \quad f_2(\boldsymbol{x}) = h(f_1(\boldsymbol{x})) = \sigma(\boldsymbol{w}_2 \odot f_1(\boldsymbol{x}), \boldsymbol{b}_2), \tag{1}$$

where $\boldsymbol{w}_1, \boldsymbol{w}_2, \boldsymbol{b}_1, \boldsymbol{b}_2 \in \mathbb{R}^d$ are the trainable weights and trainable biases, and $\sigma(\cdot, \cdot)$ represents the symmetrized ReLu activation function, defined as $\sigma(a, b) = \text{ReLu}(a - b) - \text{ReLu}(-a - b)$, applied

element-wise. In this model, each coordinate in the input is processed independently without any cross-coordinate connections. This design not only simplifies our analysis but also aligns with our definition where the features at all coordinates are independent. Because of this definition, there is no motivation for even a fully connected model to combine the features. Moreover, this model is sufficient for characterizing the feature selection and weighting processes described in the previous section, enabling us to understand the key aspects clearly. Our results derived with this model also extend to fully connected ReLU networks, as we will empirically demonstrate in Section 5.1.

We train the model using gradient flow to minimize the contrastive loss $\mathcal{L}_{CL}(f_2)$. Interestingly, in the following theorem, where we compare the pre-projection and post-projection representations, we will observe that feature 2, which is discouraged from being learned during the pretraining process, has zero weight post-projection but a non-zero weight pre-projection.

**Theorem 3.8.** *If at initialization $\boldsymbol{w}_1^{(0)} = [w_{11}^{(0)} \ w_{12}^{(0)} \dots]^\top, \boldsymbol{w}_2^{(0)} = [w_{21}^{(0)} \ w_{22}^{(0)} \dots]^\top, \boldsymbol{b}_1^{(0)} = \boldsymbol{b}_2^{(0)} = [b^{(0)} \ b^{(0)} \dots]^\top$ with $|w_{22}^{(0)}| \le \sqrt{b_0}$ and $|w_{22}^{(0)}|(|w_{12}^{(0)}| - b_0) \ge b_0$, then as $t \to \infty$, $\|f_2(\boldsymbol{e}_2)\| \to 0$, $\|f_1(\boldsymbol{e}_2)\| \ge \sqrt{b_0}$.*

The theorem indicates that pre-projection representations are more transferable. The 'destroyed' feature, feature 2, is weighted zero after the projection head but non-zero before the projection head. Therefore, using the first layer representations allows us to successfully learn downstream tasks where feature 2 is the downstream relevant feature, whereas we can't do so with the second layer representations. In practice, augmentations are imperfect and may inadvertently distort important features. Additionally, there is no one-size-fits-all augmentation suitable for all downstream tasks. With non-linear models, the projection head can save us from losing valuable information.

We note that the above advantage of the pre-projection representation diminishes when we use weight decay. Indeed, we can show that $\|f_1(\boldsymbol{e}_2)\| \to 0$ in this case as weight decay tends to shrink the weights considered unnecessary for minimizing the loss. However, we note that in practical scenarios, weight decay values are often small, e.g., $10^{-6}$, and training epochs are finite, e.g., 100, as the default setting in Chen et al. (2020). Therefore, we can estimate the effect as approximately $(1-10^{-6})^{100} \approx 1-10^{-4}$ for a weight that receives zero gradient from other sources, indicating a limited effect. Consequently, the benefit of pre-projection representations remains evident with reasonable weight decay. However, using excessive weight decay will essentially remove this benefit, as shown empirically in Section 5.1.

## 4 How the idea of projection head can benefit supervised learning

The idea of projection head has been adopted by supervised contrastive learning (Khosla et al., 2020), yet its effects have not been systematically studied nor theoretically explored. Here, we provide theoretical insights into its benefits in both supervised CL (SCL) and standard supervised learning (SL).

### 4.1 The general insights from linear models

The insights gained from Section 3.1 can be carried over to understand the case of supervised learning as well. The conclusion in Theorem 3.4 actually holds in more generality. In fact, when employing a multi-layer linear model represented as $f(\boldsymbol{x}) = \boldsymbol{W}_L\boldsymbol{W}_{L-1}\dots\boldsymbol{W}_1\boldsymbol{x}$ to minimize any loss function via gradient flow with initialization $\boldsymbol{W}_l(0)^\top\boldsymbol{W}_l(0) = \boldsymbol{W}_{l+1}(0)\boldsymbol{W}_{l+1}(0)^\top$, one can derive a conclusion in the form of $\boldsymbol{W}_l(t)^\top\boldsymbol{W}_l(t) = \boldsymbol{W}_{l+1}(t)\boldsymbol{W}_{l+1}(t)^\top$, the proof of this relationship can be found in Arora et al. (2018). The singular values of the matrix represented by the model grow exponentially as we go deeper, making the model more specialized toward the pretraining task. Therefore, when considering cutting off at lower layers, it leads to more generalizable and less specialized representations. This is particularly beneficial when the target matrix $\boldsymbol{W}^*$, determined by the training data to be represented by the model using $\boldsymbol{W}_L\dots\boldsymbol{W}_1$, doesn't adequately weight features that are important for downstream tasks.

### 4.2 Alleviating class collapse and neural collapse in non-linear models

When considering representations' transferability, a crucial aspect is their utility for finer-grained downstream tasks (e.g., distinguishing dog breeds) compared to the pretraining task (e.g., distinguishing dogs from cats). In both SCL and standard supervised learning, a common challenge arises, where representations within each class become indistinguishable at a finer-grained level. This is known as class collapse in SCL (Chen et al., 2022; Graf et al., 2021; Xue et al., 2023) and neural collapse in standard SL Papyan et al. (2020); Han et al. (2021); Zhu et al. (2021); Zhou et al. (2022b;a); Lu & Steinerberger (2022); Fang et al. (2021); Hui et al. (2022).

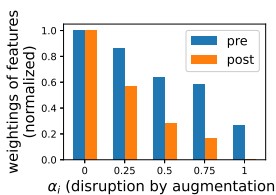 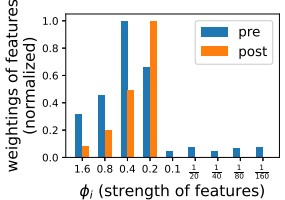 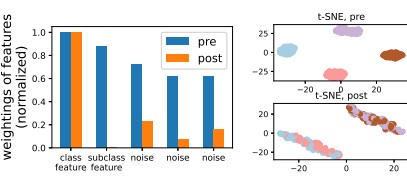

Figure 1: Weights of features in a two-layer fully connected ReLU network trained with CL. **Left**: With all features having equal strength, those that are more disrupted by augmentation have smaller/zero weights. **Right**: With augmentation treating all features equally. features with the largest/smallest strength are weighted less compared to those with intermediate strength. In both **right** and **left**, weights are more equal pre-projection.

Figure 2: **Left:** Weights of features in a two-layer fully connected ReLU network trained using SCL. The subclass feature is not represented post-projection but is represented pre-projection. **Right:** As a result, the four subclasses are only separable in pre-projection representations.

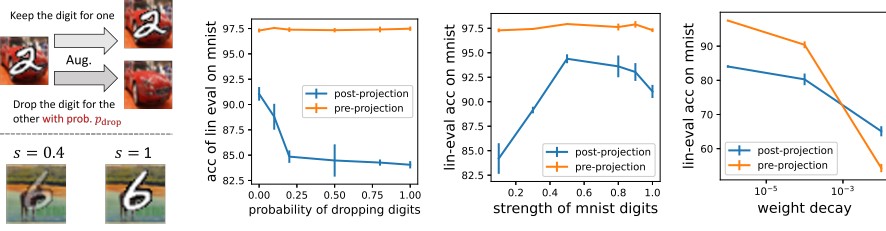

(a) An illustration    (b) Effect of data aug.    (c) Effect of strength    (d) Effect of wd

Figure 3: Results on MNIST-on-CIFAR-10. (a) The data augmentation keeps the digit for one image in the positive pair and randomly drops the digit for the other with a probability $p_{\text{drop}}$. Pre-projection is more beneficial (b) with more inappropriate augmentation (large $p_{\text{drop}}$) during pretraining, (c) when digits are very weak/strong during pretraining, and (d) when weight decay is smaller.

Interestingly, we show that lower layers can learn subclass-level features not represented in the final layer in supervised learning, suggesting that using and then discarding the projection head can mitigate the issue of class/neural collapse. To theoretically characterize this, we consider the following data distribution.

**Definition 4.1.** *We have four subclasses represented by* $(y, y_{sub})$*, where* $y \in \{1, -1\}$ *is the class label, and* $y_{sub} \in \{1, -1\}$*. For each input vector* $\boldsymbol{x} \in \mathbb{R}^d$ *in subclass* $(y, y_{sub})$*,* $\boldsymbol{x} = [y \ y_{sub} \ \ldots]^\top$ *with the other coordinates being independent and symmetrically distributed.*

We consider training the non-linear diagonal network defined in Equation 1 on the above data distribution. For supervised CL, we consider the following spectral loss, which a natural generalization from that for self-supervised ($\mathcal{L}_{CL}$) (HaoChen et al., 2021; Xue et al., 2022; Saunshi et al., 2022; HaoChen & Ma, 2022; Garrido et al., 2022; Xue et al., 2023).

$$\mathcal{L}_{SCL}(f_2) = -2\mathbb{E}_{\boldsymbol{x},\boldsymbol{x}^+ \sim \text{ the same class}}[f_2(\boldsymbol{x})^\top f_2(\boldsymbol{x}^+)] + \mathbb{E}_{\boldsymbol{x},\boldsymbol{x}^- \text{independently drawn}}[(f_2(\boldsymbol{x})^\top f_2(\boldsymbol{x}^-))^2].$$

For standard supervised learning, we consider the Mean Squared Error (MSE) loss. Since the network's output $f_2(\boldsymbol{x})$ is $d$-dimensional, we map it to a scalar by adding a linear layer with all weights set to 1. Let $\mathbf{1} \in \mathbb{R}^d$ be a vector with all elements equal to 1, then the loss can be written as:

$$\mathcal{L}_{SL}(f_2) = \mathbb{E}_{(\boldsymbol{x},y)}(f_2(\boldsymbol{x})^\top \mathbf{1} - y)^2.$$

According to Definition 4.1, we can use $\|f_i(\boldsymbol{e}_1)\|$ and $\|f_i(\boldsymbol{e}_2)\|$ to determine the model's weighting of the class feature and the subclass feature, respectively, in the $i$-th layer representations. Next theorem demonstrates that the features indicating subclasses are learned pre-projection but not post-projection.

**Theorem 4.2.** *Under the same assumption about initialization as in Theorem 3.8, and* $w_{12}^{(0)} w_{22}^{(0)} > 0$*, for either the model trained to minimize* $\mathcal{L}_{SCL}(f_2)$ *or* $\mathcal{L}_{SL}(f_2)$ *with gradient flow, as* $t \to \infty$*,* $\|f_2(\boldsymbol{e}_2)\| \to 0$*,* $\|f_1(\boldsymbol{e}_2)\| \geq \sqrt{b_0}$*.*

## 5 EXPERIMENTS

### 5.1 CONTROLLED EXPERIMENTS ON SELF-SUPERVISED CL

The empirical benefits of the projection head are already well-established, and evidence can be found in numerous large-scale experiments. Instead, in this subsection, we focus on controlled

proof-of-concept experiments, allowing us to see the underlying mechanisms that are not shown in existing works and validate our theoretical results from Section 3.

**Synthetic data.** We train a two-layer ReLU network on data drawn from the distribution defined in Definition 3.1, with the second ReLU layer serving as the projection head. The network is randomly initialized. More details are in Appendix C.1. We conduct experiments in two settings. (1) In setting 1, we let all five features have equal strength and let the augmentation disrupts features differently, with feature 1 perfectly preserved and feature 5 completely randomized. Figure 1 left shows the weights assigned to features at pre- and post-projection. Consistent with the conclusion from Theorem 3.5, features disrupted more by data augmentation are assigned smaller weights. However, these features are weighted more equally pre-projection. Notably, feature 5 has zero weight post-projection but non-zero weights pre-projection, aligning with the finding in Theorem 3.8. (2) In setting 2, we aim to demonstrate the moderating effect reflected in Theorem 3.5, We let $\alpha_i$'s be equal, controlling the factor of data augmentation, then plot the weights of features with different strength in Figure 1 right. Features being either too strong or too weak are weighted less than intermediate strength features, but in pre-projection, they are less underweighted compared to post-projection.

**MNIST-on-CIFAR-10.** We design the following dataset for pretraining, with the downstream task on the original MNIST. The *pretraining dataset* is based on CIFAR-10, where each image combines a CIFAR-10 image with a MNIST digit. Specifically, the pixel values within the digit area are calculated as $(1-s)\times$ CIFAR-10 image $+s\times$ MNIST digit, while the values outside this area remain unchanged as in the CIFAR-10 image. Here $s$ controls the strength of the downstream-relevant feature (MNIST digits) during pretraining. We use the following *data augmentation* during pretraining. For each positive pair, we keep one image's digit, while dropping the other's digit with probability $p_{drop}$. A larger $p_{drop}$ means that the augmentation disrupts the downstream-relevant feature more. See Figure 3a for an illustration. The *downstream task* is classification on the original MNIST dataset. We feed MNIST digits to the pretrained model and train a linear classifier on representations. We pretrain ResNet-18s with one-hidden-layer MLP as the projection head using the popular SimCLR loss (Chen et al., 2020) with varying $s$ and $p_{drop}$ and report the evaluation results in Figure 3.

Figure 3b shows the downstream accuracy against $p_{\text{drop}}$ with $s = 1$. Pre-projection representations outperform post-projection representations, and the gap widens as $p_{\text{drop}}$ increases. This confirms that inappropriate augmentations can significantly amplify the superiority of pre-projection representations, aligning with Corollary 3.7. Figure 3c shows the downstream accuracy against $s$, the strength of digits, with $p_{\text{drop}} = 0$. While the pre-projection accuracy barely changes, the post-projection accuracy increases and decreases, as $s$ increases, making the benefit of using pre-projection more pronounced when either the digit is too weak or too strong, aligning with Corollary 3.7. Figure 3d with $p_{\text{drop}} = 1$ demonstrates the impact of weight decay. Larger weight decay diminishes the benefits of pre-projection representations, aligning with our discussion in Section 3.2, and can even turn them into a detriment. However, pre-projection remains superior with reasonable weight decay.

In Appendix D.1, we present two additional experiments involving minimally modified CIFAR-10 images to further support our conclusions. We also discuss the effect of early stopping in Appendix D.2.

## 5.2 Projection Head in Supervised Learning

We present the following experiments showing broader implications of projection head in SCL and SL.

**Coarse-to-fine transferability on synthetic data.** We consider the data distribution in Definition 4.1. A two-layer ReLU network is trained from random initialization on such data. Figure 2 visualizes the weight assigned to each input component by the model (left) and the representations (right) at each layer. Consistent with the conclusion from Theorem 4.2, we observe that the subclass feature is assigned a weight of zero post-projection but has a non-zero weight pre-projection. As a result, pre-projection representations do not suffer from class collapse, making them more transferable.

**Coarse-to-fine transferability on CIFAR-100.** We pretrain a ResNet-18s on CIFAR-100 with 20 coarse-grained labels and conducted linear evaluation on representations using both these 20 labels and 100 fine-grained labels, separately (details in Appendix C.1). The results

Table 1: The pre-projection reps. suffer less from class/neural collapse, allowing for better fine-grained classification.

|  | coarse | | fine | |
|---|---|---|---|---|
|  | pre | post | pre | post |
| SCL | $55.2_{\pm0.4}$ | $53.6_{\pm0.1}$ | $36.0_{\pm0.1}$ | $25.7_{\pm0.7}$ |
| SL | $53.1_{\pm0.3}$ | $53.2_{\pm0.7}$ | $33.7_{\pm0.3}$ | $29.7_{\pm0.3}$ |

Table 2: The fixed reweighting head can yield improvements comparable to those of the trainable projection head. For the distribution shift scenario, we report the average test accuracies for 2, 4, 8, 16, 32, 64, and 128 shots adaption. More detaied results are in Appendix C.2.

| Scenario | Dataset | Alg. | Performance Measure | Performance | | |
|---|---|---|---|---|---|---|
| | | | | vanilla | proj | reweight |
| synthetic | M-on-C | SSCL | digit clf. acc. | 77.0 | 97.3 | 97.3 |
| coarse-to-fine | CIFAR100 | SCL | fine-grained clf. acc. | 21.8 | 36.0 | 30.2 |
| coarse-to-fine | CIFAR100 | SL | fine-grained clf. acc. | 31.44 | 33.7 | 32.2 |
| distribution shift | UrbanCars | SL | few-shot adaption acc. | 82.2 | 86.1 | 87.0 |

are compared in Table 1. Pre-projection representations yield higher accuracy in general, but with the benefit much more significant in the fine-grained downstream task. This emphasizes that pre-projection representations are less susceptible to class/neural collapse, resulting in more distinguishable representations within each pretraining class.

**Few-shot adaption to distribution shift on UrbanCars.** Subpopulation shift typically involves scenarios where some data groups are underrepresented in the source domain but well-represented in the target domain. Recent works have demonstrated the effectiveness of retraining the final linear layer using target data to adapt to distribution shifts Rosenfeld et al. (2022); Kirichenko et al. (2022); Mehta et al. (2022). However, it has been shown that this approach leads to suboptimal results when the target data for training the linear layer is scarce Chen et al. (2023). This could be attributed to the suboptimal quality of penultimate representations. Here, in Figure 4, we demonstrate that by applying the projection head technique and performing adaption on pre-projection representations can lead to better performance. The details are provided in the Appendix C.1. We also present experiments on shifted ImageNets in Appendix D.3.

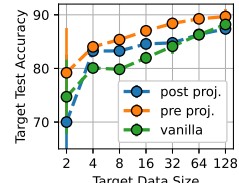

Figure 4: Performance of few-shot adaption.

## 6 REPLACING THE PROJECTION HEAD WITH A FIXED REWEIGHTING HEAD

Based on the observations above, we now explore an alternative design that employs a fixed reweighting head instead of a trainable projection head. Let $\mathbf{r} = [r_1, \ldots, r_p]^\top$ be the representation output by the encoder. The reweighting head $h_{\mathrm{rw}}()$ acts as $h_{\mathrm{rw}}(\boldsymbol{r}) = [r_1, \frac{1}{\kappa} r_2, \ldots, \frac{1}{\kappa^{p-1}} r_p]^\top$, where $\kappa > 1$ is a hyperparameter. This mimics the role of the projection head described earlier. During pretraining, the reweighting head unequally weight features by assigning different weights to the dimensions of representations, with $\kappa$ controlling how unequal they are. This allows the encoder before the head to weight input features more equally. Note that with a slightly large $\kappa$, dimensions at the end will be almost 'turned off' after the head, which can result in representations after the head representing fewer features than before. This mimics the effect discussed for non-linear models in Sections 3 and 4. We evaluate this approach in the scenarios from Sec. 5.2 and demonstrate that the fixed reweighting head can achieve improvements comparable to those of the projection head, as shown in Table 2. This has two implications: (1) it serves as further evidence for our theoretical conclusions in Sections 3 and 4, indicating that the reweighting effect accounts for all or most of the improvements achieved by the projection head. (2) it opens up possibilities for designing more straightforward and interpretable alternatives to the projection head. It would be interesting for future work to explore the effects of making $\kappa$ trainable rather than fixed.

## 7 CONCLUSION

We provided the first theoretically rigorous explanation for the intriguing empirical success of using the projection head for self-supervised contrastive learning, supervised contrastive learning, and supervised learning. We demonstrated that lower layers represent features more evenly in linear networks and can represent more features in non-linear networks. This enhances the generalizability and transferability of the representations, especially when downstream-relevant features are weak in the pretraining data or heavily distorted by data augmentations. Interestingly, we also show the benefits when the downstream-relevant features are prominent in the pretraining data. We validated our theoretical findings through extensive experiments on both synthetic and real-world data. Finally, we demonstrate how a fixed reweighting head can achieve performance comparable to the projection head, providing further evidence to support our theoretical conclusions. We also hope that this will offer valuable guidance for future design choices.

**Acknowledgements** This research is partially supported by the National Science Foundation CAREER Award 2146492.

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

## A  THEORETICAL ANALYSIS

### A.1  STRUCTURE OF WEIGHTS

**Theorem A.1** (Weights of the minimum norm minimizer). *The global minimizer of the CL loss $\mathcal{L}_{CL}$ with the smallest norm, defined as $\|W_1^\top W_1\|_F^2 + \|W_2^\top W_2\|_F^2$, satisfies*

$$W_1 W_1^\top = W_2^\top W_2.$$

*Proof.* Observe that

$$
\begin{aligned}
\|W_1^\top W_1\|_F^2 + \|W_2^\top W_2\|_F^2 &= \|W_1 W_1^\top\|_F^2 + \|W_2^\top W_2\|_F^2 \\
&= \operatorname{Tr}((W_1 W_1^\top)^2 + (W_2^\top W_2)^2) \\
&= \operatorname{Tr}((W_1 W_1^\top - W_2^\top W_2)^2 - W_1 W_1^\top W_2^\top W_2 - W_2^\top W_2 W_1 W_1^\top) \\
&= \operatorname{Tr}((W_1 W_1^\top - W_2^\top W_2)^2) - 2\operatorname{Tr}(W_1^\top W_2^\top W_2 W_1)
\end{aligned}
$$

Consider any value of $W = W_2 W_1$ that is a global minimizer of the CL loss. This determines the value of $\operatorname{Tr}(W_1^\top W_2^\top W_2 W_1)$. Hence the norm is minimized when $\operatorname{Tr}((W_1 W_1^\top - W_2^\top W_2)^2) = 0$. But $(W_1 W_1^\top - W_2^\top W_2)^2$ is positive definite so this occurs if and only if $W_1 W_1^\top = W_2^\top W_2$.

On the other hand, this minimum is achievable, since if $W$ has SVD $W = U\Sigma V^\top$, then $W_1 = \Sigma^{\frac{1}{2}} V^\top, W_2 = U\Sigma^{\frac{1}{2}}$ is a solution satisfying $W_1 W_1^\top = W_2^\top W_2$. $\qquad\square$

**Theorem A.2** (Weights of the model trained with gradient flow, proved in Arora et al. (2018)). *Suppose the initialization satisfies*

$$W_1(0)W_1^\top(0) = W_2^\top(0)W_2(0).$$

*Using gradient flow, at any time $t$, we have*

$$W_1(t)W_1^\top(t) = W_2^\top(t)W_2(t).$$

*Proof.* We reiterate the proof. Let $Z = f(X)$ be the matrix of model outputs. Calculate that the gradients are

$$
\frac{\partial \mathcal{L}}{\partial W_1} = W_2^\top \frac{\partial \mathcal{L}}{\partial Z} X^\top
$$

$$
\frac{\partial \mathcal{L}}{\partial W_2} = \frac{\partial \mathcal{L}}{\partial Z} X^\top W_1^\top
$$

In addition, the chain rule and gradient flow gives

$$
\frac{d}{dt}(W_1 W_1^\top) = -W_1 \left(\frac{\partial \mathcal{L}}{\partial W_1}\right)^\top - \frac{\partial \mathcal{L}}{\partial W_1} W_1^\top
$$

$$
\frac{d}{dt}(W_2^\top W_2) = -\left(\frac{\partial \mathcal{L}}{\partial W_2}\right)^\top W_2 - W_2^\top \frac{\partial \mathcal{L}}{\partial W_2}.
$$

Substituting, we see that $\frac{d}{dt}(W_1 W_1^\top) = \frac{d}{dt}(W_2^\top W_2)$. The conclusion follows. $\qquad\square$

### A.2  WEIGHTS OF FEATURES

The following analysis is performed for self-supervised CL loss, under the condition that $W_1 W_1^\top = W_2^\top W_2$.

Define the covariance of augmented examples

$$M := \mathbb{E}[\mathcal{A}(x)(\mathcal{A}(x))^\top] = \mathbf{Diag}([\phi_1^2 + \sigma^2 \ldots \phi_d^2 + \sigma^2]), \tag{2}$$

and the covariance of augmentation centers

$$\tilde{M} := \mathbb{E}[\mathbb{E}\mathcal{A}(x)(\mathbb{E}\mathcal{A}(x))^\top] = \mathbf{Diag}([(1 - 2\alpha_1)^2 \phi_1^2 \ldots (1 - 2\alpha_d)^2 \phi_d^2]). \tag{3}$$

Let $\boldsymbol{W} = \boldsymbol{W}_2 \boldsymbol{W}_1$. By invoking Lemma B.2 in Xue et al. (2023) about the minimizer of the loss, given the fact that $\boldsymbol{M}, \tilde{\boldsymbol{M}}$ are full rank, we have

$$\boldsymbol{W}^\top \boldsymbol{W} \boldsymbol{M} = [\boldsymbol{M}^{-1} \tilde{\boldsymbol{M}}]_p, \tag{4}$$

where notation $[\boldsymbol{A}]_p$ represents the matrix composed of the first $p$ eigenvalues and eigenvectors of a positive semidefinite $\boldsymbol{A}$ (if $p \geq \text{rank}\boldsymbol{A}$ then $[\boldsymbol{A}]_p = \boldsymbol{A}$). Therefore, substituting yields

$$\boldsymbol{W}^\top \boldsymbol{W} = [\textbf{Diag}([\frac{(1-2\alpha_1)^2 \phi_1^2}{\phi_1^2 + \sigma^2} \dots])]_p \textbf{Diag}([\frac{1}{\phi_1^2 + \sigma^2} \dots]) := \boldsymbol{D} \tag{5}$$

Note that $[\textbf{Diag}([\frac{(1-2\alpha_1)^2 \phi_1^2}{\phi_1^2 + \sigma^2} \dots])]_p$ is a matrix that only keeps the $p$ largest diagonal entries of $\textbf{Diag}([\frac{(1-2\alpha_1)^2 \phi_1^2}{\phi_1^2 + \sigma^2} \dots])$.

To avoid cluttered notations, below we consider the case where $p \leq d$. However, we note that similar analysis holds for $p > d$. Now we can obtain $\boldsymbol{W} = \boldsymbol{U}\sqrt{\boldsymbol{D}}$, where $\boldsymbol{U} \in \mathbb{R}^{p \times p}$ is a matrix with orthonormal rows. Then

$$\|f_2(\boldsymbol{e}_i)\| = \|\boldsymbol{W}\boldsymbol{e}_i\| = \|\boldsymbol{U}[i, :]\sqrt{\boldsymbol{D}}[i, i]\| = \sqrt{\boldsymbol{D}}[i, i], \tag{6}$$

which yields the conclusion about $\|f_2(\boldsymbol{e}_i)\|$ in Theorem 3.5.

Now let's examine $\|f_1(\boldsymbol{e}_i)\|$. Let $\boldsymbol{U}_1 \boldsymbol{S}_1 \boldsymbol{V}_1^\top$ and $\boldsymbol{U}_2 \boldsymbol{S}_2 \boldsymbol{V}_2^\top$ be $\boldsymbol{W}_1$ and $\boldsymbol{W}_2$'s SVD, respectively. Given that $\boldsymbol{W}_1 \boldsymbol{W}_1^\top = \boldsymbol{W}_2^\top \boldsymbol{W}_2$, we have

$$\boldsymbol{U}_1 (\boldsymbol{S}_1 \boldsymbol{S}_1^\top) \boldsymbol{U}_1^\top = \boldsymbol{V}_2 \boldsymbol{S}_2^2 \boldsymbol{V}_2^\top, \tag{7}$$

implying that the non-zero singular values of $\boldsymbol{W}_1$ and $\boldsymbol{W}_2$ matches, and those singular values corresponding columns in $\boldsymbol{U}_1$ match those in $\boldsymbol{V}_2$. Given that $\boldsymbol{W}_2 \boldsymbol{W}_1 = \boldsymbol{W} = \boldsymbol{U}\sqrt{\boldsymbol{D}}$, we have

$$\begin{aligned}
\boldsymbol{W}_2 \boldsymbol{W}_1 =& \boldsymbol{U}_2 \boldsymbol{S}_2 \boldsymbol{S}_1 \boldsymbol{V}_1^\top = \boldsymbol{U}\sqrt{\boldsymbol{D}} && \text{because singular vectors match} \\
\boldsymbol{S}_2 \boldsymbol{S}_1 \boldsymbol{V}_1^\top =& \boldsymbol{U}_2^\top \boldsymbol{U}\sqrt{\boldsymbol{D}} && \text{because } \boldsymbol{U}_2 \in \mathbb{R}^{p \times p} \text{ is unitary.} \\
\boldsymbol{S}_1 \boldsymbol{V}_1^\top =& \boldsymbol{U}_2^\top \boldsymbol{U}\sqrt[4]{\boldsymbol{D}} && \text{because singular values match.}
\end{aligned} \tag{8}$$

Given equation 8, we have

$$\begin{aligned}
\|f_1(\boldsymbol{e}_i)\| =& \|\boldsymbol{W}_1 \boldsymbol{e}_i\| \\
=& \|\boldsymbol{U}_1 \boldsymbol{S}_1 \boldsymbol{V}_1^\top \boldsymbol{e}_i\| \\
=& \|\boldsymbol{S}_1 \boldsymbol{V}_1^\top \boldsymbol{e}_i\| \\
=& \|\boldsymbol{U}_2^\top \boldsymbol{U}\sqrt[4]{\boldsymbol{D}}\boldsymbol{e}_i\| \\
=& \sqrt[4]{\boldsymbol{D}}[i, i],
\end{aligned}$$

which completes the proof of Theorem 3.5.

### A.3 COMPARING SAMPLE COMPLEXITY

**Definition A.3.** *We say that a data distribution $\mathcal{P}$ is separable with a $(\gamma, \rho)$ margin if where exists $\boldsymbol{w}^*, b^*$ s.t. $\|\boldsymbol{w}^*\| = 1$ and*

$$\mathcal{P}(\{(\boldsymbol{x}, y) : \|\boldsymbol{x}\| \leq \rho \wedge y(\boldsymbol{w}^{*\top}\boldsymbol{x} + b^*) \geq \gamma\}) = 1).$$

It is well-known that the sample complexity of hard-margin SVM only grows with $r := (\rho/\gamma)^2$ (e.g., Bartlett & Shawe-Taylor (1999) ). Therefore, we refer to $r$ as the sample complexity indicator. From the analysis in Section A.2, we know that

$$f_1(\boldsymbol{e}_i) = \boldsymbol{U}_1 \boldsymbol{U}_2 \boldsymbol{U}\sqrt[4]{\boldsymbol{D}}\boldsymbol{e}_i = \sqrt[4]{\boldsymbol{D}}[i, i](\boldsymbol{U}_1 \boldsymbol{U}_2 \boldsymbol{U})[i, :].$$

Since $\boldsymbol{U}_1 \boldsymbol{U}_2$ is unitary and $\boldsymbol{U}$ has orthornormal rows, $\boldsymbol{U}_1 \boldsymbol{U}_2 \boldsymbol{U}$ also has orthornormal rows. Thus, $f_1(\boldsymbol{e}_i)$'s are orthonormal. The same conclusion holds for $f_i(\boldsymbol{e}_i)$'s as well.

Now, given Definition A.3, by letting $\boldsymbol{w}^* = \frac{f_i(\boldsymbol{e}_{j^*})}{\|f_i(\boldsymbol{e}_{j^*})\|}$ and $b^* = 0$, we can obtain $\gamma = \|f_i(\boldsymbol{e}_{j^*})\|\hat{\phi}_{j^*}$ for the $i$-th layer's representations. We also have $p = \sqrt{\sum_{j=1}^p \|f_i(\boldsymbol{e}_j)\|^2 \hat{\phi}_j^2}$. Thus, the sample complexity indicator is $r_i = (\frac{\sqrt{\sum_{j=1}^p \|f_i(\boldsymbol{e}_j)\|^2 \hat{\phi}_j^2}}{\|f_i(\boldsymbol{e}_{j^*})\|\hat{\phi}_{j^*}})^2$, for $i = 1, 2$. Substituting the values of $\|f_i(\boldsymbol{e}_j)\|$'s into the comparison between $r_1$ and $r_2$, with some algebraic manipulation yields Theorem 3.6.

## A.4 ANALYSIS FOR NON-LINEAR MODELS

We introduce the following two lemmas which allow us to analyze coordinates of the model separately.

**Lemma A.4.** *Suppose the model $\boldsymbol{f}$ can be decomposed coordinate-wise $f_1, \ldots, f_p$ and each $f_i$ is odd. Also suppose the dataset $\mathcal{D}$ follows a coordinate wise symmetric distribution, namely the pdf $p$ satisfies*

$$p(x_1, \ldots, x_{i-1}, x_i, x_{i+1}, \ldots x_p) = p(x_1, \ldots, x_{i-1}, -x_i, x_{i+1}, \ldots x_p) \tag{9}$$

*for any $i$. Then the contrastive loss can be decomposed coordinate-wise*

$$\mathcal{L} = \sum_{i=1}^{p} -2\mathbb{E}[f_i(x_i)f_i(x_i^+)] + \mathbb{E}\left[\left(f_i(x_i)f_i(x_i^-)\right)^2\right] \tag{10}$$

*Proof.*

$$\mathcal{L} = -2\mathbb{E}[\boldsymbol{f}(\boldsymbol{x})^\top \boldsymbol{f}(\boldsymbol{x})_+] + \mathbb{E}[(\boldsymbol{f}(\boldsymbol{x})^\top \boldsymbol{f}(\boldsymbol{x})_-)^2] \tag{11}$$

$$= -2\mathbb{E}\left[\sum_{i=1}^{p} f_i(x_i)f_i(x_i^+)\right] + \mathbb{E}\left[\left(\sum_{i=1}^{p} f_i(x_i)f_i(x_i^-)\right)^2\right] \tag{12}$$

$$= -2\mathbb{E}\left[\sum_{i=1}^{p} f_i(x_i)f_i(x_i^+)\right] + \mathbb{E}\left[\mathbb{E}_{\sigma_i \sim Unif\{-1,1\}}\left[\left(\sum_{i=1}^{p} f_i(\sigma_i x_i)f_i(x_i^-)\right)^2\right]\right] \tag{13}$$

$$= -2\mathbb{E}\left[\sum_{i=1}^{p} f_i(x_i)f_i(x_i^+)\right] + \mathbb{E}\left[\mathbb{E}_{\sigma_i \sim Unif\{-1,1\}}\left[\left(\sum_{i=1}^{p} \sigma_i f_i(x_i)f_i(x_i^-)\right)^2\right]\right] \tag{14}$$

$$= -2\mathbb{E}\left[\sum_{i=1}^{p} f_i(x_i)f_i(x_i^+)\right] + \mathbb{E}\left[\sum_{i=1}^{p} \left(f_i(x_i)f_i(x_i^-)\right)^2\right] \tag{15}$$

$$= \sum_{i=1}^{p} -2\mathbb{E}[f_i(x_i)f_i(x_i^+)] + \mathbb{E}\left[\left(f_i(x_i)f_i(x_i^-)\right)^2\right] \tag{16}$$

Note that this hold for both supervised and unsupervised contrastive loss, since they only differ in how positive pairs are defined. □

**Lemma A.5.** *Under the same assumptions as A.4, MSE loss as defined in 2 can be decomposed coordinate-wise.*

*Proof.*

$$\mathcal{L} = \mathbb{E}_{(\boldsymbol{x},y)}\left[(\boldsymbol{f}(\boldsymbol{x})^\top \mathbf{1} - y)^2\right] \tag{17}$$

$$= \mathbb{E}_{(\boldsymbol{x},y)}\left[\left(-y + \sum_{i=1}^{p} f_i(x_i)\right)^2\right] \tag{18}$$

$$= \mathbb{E}_{(\boldsymbol{x},y)}\left[y^2 - 2\sum_{i=1}^{p} yf_i(x_i) + \mathbb{E}_{\sigma_i \sim Unif\{-1,1\}}\left[\left(\sum_{i=1}^{p} f_i(\sigma_i x_i)\right)^2\right]\right] \tag{19}$$

$$= \mathbb{E}_{(\boldsymbol{x},y)}\left[y^2 - 2\sum_{i=1}^{p} yf_i(x_i) + \mathbb{E}_{\sigma_i \sim Unif\{-1,1\}}\left[\left(\sum_{i=1}^{p} \sigma_i f_i(x_i)\right)^2\right]\right] \tag{20}$$

$$= \mathbb{E}_{(\boldsymbol{x},y)}\left[y^2 - 2\sum_{i=1}^{p} yf_i(x_i) + \sum_{i=1}^{p} (f_i(x_i))^2\right] \tag{21}$$

$$= (1-p)\mathbb{E}_y[y^2] + \sum_{i=1}^{p} \mathbb{E}_{(\boldsymbol{x},y)}\left[(f_i(x_i) - y)^2\right] \tag{22}$$

$\square$

It is easy to check that the setting with the diagonal network and given data distribution satisfies the conditions of the lemma.

By the above lemmas, we can consider optimizing over each coordinate separately.

**Theorem A.6** (Contrastive Loss). *Assume* $|w_{22}^{(0)}| \leq \sqrt{b^{(0)}}$ *and* $|w_{22}^{(0)}|(|w_{12}^{(0)}| - b^{(0)}) \geq b^{(0)}$, *then as* $t \to \infty$, $|f_2(\boldsymbol{e}_2)| \to 0$ *and* $|f_1(\boldsymbol{e}_2)| \geq \sqrt{b^{(0)}}$.

*Proof.* Let $\boldsymbol{f}_2(\boldsymbol{x}) = (z_1, \ldots, z_n)$ be the embeddings outputted by the second layer. Then the coordinate-wise decomposition of the contrastive loss takes the form

$$\mathcal{L} = \mathbb{E}_{z \sim f(\mathcal{D})}[(z_1^2 - 1)^2 + z_2^4 + \cdots + z_n^4] \tag{23}$$

From here it is clear that the an optimal solution maps the second coordinate of every embedding to zero. Writing down the gradients for the weights and threshold,

$$\frac{\partial L}{\partial w_{22}} = \mathbb{E}_{\boldsymbol{x} \sim \mathcal{D}}[4z_2^3 \sigma'(w_{22}\sigma(w_{12}x_2, b_{12}), b_{22})\sigma(w_{12}x_2, b_{12})]$$

$$\frac{\partial L}{\partial w_{12}} = \mathbb{E}_{\boldsymbol{x} \sim \mathcal{D}}[4z_2^3 \sigma'(w_{22}\sigma(w_{12}x_2, b_{12}), b_{22})w_{22}\sigma'(w_{12}x_2)x_2]$$

$$\frac{\partial L}{\partial b_{22}} = \mathbb{E}_{\boldsymbol{x} \sim \mathcal{D}}[-4z_2^3 \sigma'(w_{22}\sigma(w_{12}x_2, b_{12}), b_{22})]$$

$$\frac{\partial L}{\partial b_{12}} = \mathbb{E}_{\boldsymbol{x} \sim \mathcal{D}}[-4z_2^3 \sigma'(w_{22}\sigma(w_{12}x_2, b_{12}), b_{22})w_{22}]$$

Observe that $|w_{12}|$ and $|w_{22}|$ are both decreasing and $b_{12}, b_{22}$ are both increasing throughout training, so that as $t \to \infty$, the second coordinate of all embeddings goes to zero. Namely, this means that

$$|w_{22}^{(t)}| \left(|w_{12}^{(t)}| - b_{12}^{(t)}\right) \leq b_{22}^{(t)} \tag{24}$$

But since weights are decreasing and thresholds are increasing, $|w_{22}^{(t)}| \leq \sqrt{b^{(0)}} \leq \sqrt{b_{22}^{(t)}}$. It follows that $(|w_{12}^{(t)}| - b_{12}^{(t)}) \geq \sqrt{b_{22}^{(t)}} \geq \sqrt{b^{(0)}}$. Rearranging gives

$$|w_{12}^{(t)}| \geq \sqrt{b^{(0)}} + b_{12}^{(t)},$$

as desired. $\square$

**Theorem A.7** (MSE Loss). *Assume* $|w_{22}^{(0)}| \leq \sqrt{b^{(0)}}$ *and* $|w_{22}^{(0)}|(|w_{12}^{(0)}| - b^{(0)}) \geq b^{(0)}$, *and* $w_{22}^{(0)}$ *and* $w_{12}^{(0)}$ *have the same sign, then as* $t \to \infty$, $|f_2(\boldsymbol{e}_2)| \to 0$ *and* $|f_1(\boldsymbol{e}_2)| \geq \sqrt{b^{(0)}}$.

*Proof.* Let $\boldsymbol{f}_2(\boldsymbol{x}) = (z_1, \ldots, z_n)$ be the embeddings outputted by the second layer. Then the coordinate-wise decomposition of the loss takes the form

$$\mathcal{L} = \mathbb{E}_{z \sim f(\mathcal{D})}[(z_1 - 1)^2 + z_2^2 + \cdots + z_n^2] + C \tag{25}$$

where C is independent of the weights.

From here it is clear that the an optimal solution maps the second coordinate of every embedding to zero. W.L.O.G., assume $w_{12}^{(0)}, w_{22}^{(0)} > 0$. Writing down the gradients for the weights and threshold,

$$\frac{\partial L}{\partial w_{22}} = \mathbb{E}_{\boldsymbol{x} \sim \mathcal{D}}[2z_2 \sigma'(w_{22}\sigma(w_{12}x_2, b_{12}), b_{22})\sigma(w_{12}x_2, b_{12})]$$

$$\frac{\partial L}{\partial w_{12}} = \mathbb{E}_{\boldsymbol{x} \sim \mathcal{D}}[2z_2 \sigma'(w_{22}\sigma(w_{12}x_2, b_{12}), b_{22})w_{22}\sigma'(w_{12}x_2)x_2]$$

$$\frac{\partial L}{\partial b_{22}} = \mathbb{E}_{\boldsymbol{x} \sim \mathcal{D}}[-2z_2 \sigma'(w_{22}\sigma(w_{12}x_2, b_{12}), b_{22})]$$

$$\frac{\partial L}{\partial b_{12}} = \mathbb{E}_{\boldsymbol{x} \sim \mathcal{D}}[-2z_2 \sigma'(w_{22}\sigma(w_{12}x_2, b_{12}), b_{22})w_{22}]$$

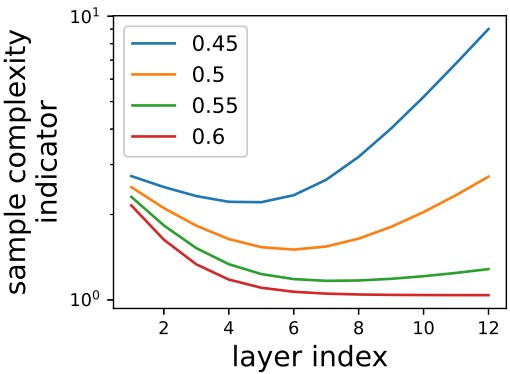

Figure 5: The value of the sample complexity indicator at different layers. Legends show the weight assigned to the downstream relevant weight by the pretrained model. We observe that the optimal layer shifts lower when the pretrained model assigns less weight to the downstream relevant features.

Observe that $w_{12}$ and $w_{22}$ are both decreasing and $b_{12}, b_{22}$ are both increasing throughout training, so that as $t \to \infty$, the second coordinate of all embeddings goes to zero. Namely, this means that

$$w_{22}^{(t)} \left( w_{12}^{(t)} - b_{12}^{(t)} \right) \leq b_{22}^{(t)} \tag{26}$$

But since weights are decreasing and thresholds are increasing, $w_{22}^{(t)} \leq \sqrt{b^{(0)}} \leq \sqrt{b_{22}^{(t)}}$. It follows that $(w_{12}^{(t)} - b_{12}^{(t)}) \geq \sqrt{b_{22}^{(t)}} \geq \sqrt{b^{(0)}}$. Rearranging gives

$$w_{12}^{(t)} \geq \sqrt{b^{(0)}} + b_{12}^{(t)},$$

as desired. □

## B  DISCUSSION ON MULTI-LAYER LINEAR MODEL

As mentioned in Section 4, Theorem 3.4 can be extended to multi-layer models, which gives us $\boldsymbol{W}_l(t)^\top \boldsymbol{W}_l(t) = \boldsymbol{W}_{l+1}(t)\boldsymbol{W}_{l+1}(t)^\top$. Based on this, going through a similar process as in Appendix A.2 and Appendix A.3, we will obtain the following expression of the sample complexity indicator for each layer $l$:

$$r_l = \frac{\sum_{j=1}^p c_j^{2l/L} \hat{\phi}_j^2}{c_{j^*}^{2l/L} \hat{\phi}_{j^*}}$$

where

$$c_j = \begin{cases} \frac{(1-\alpha_j)\phi_j}{\phi_j^2+\sigma^2}, & \text{if } j \in \{j_1, \ldots, j_{\min\{d,p\}}\} \\ 0, & \text{else} \end{cases}$$

The definitions of $j_1, \ldots, j_{\min\{d,p\}}$ remain the same as in Theorem 3.5, and other quantities are consistent with the definitions provided in Section 3. Here, $c_j$ represents the weight the full model would allocate to the $j$-th feature. Although the expression appears complex, some intuition can be gleaned from extreme scenarios: If $c_{j^*}$ is the largest, indicating that the full model assigns the most weight to the downstream relevant feature, $r_l$ decreases with $l$. This means one should just use the final-layer representations. 2. Conversely, if $c_{j^*}$ is the smallest among non-zero $c_j$'s, indicating that the full model assigns the least weight to the downstream relevant feature, $r_l$ increases with $l$. In this case, using the lowest layer would be preferable. Applying these observations in conjunction with the relationship between $c_j$ and $\alpha_j, \phi_j$, similar conclusions to those in Corollary 3.7 regarding the impact of augmentation and feature strength for multi-layer models can be drawn.

**The greater the mismatch between the pretraining and downstream tasks, the lower the optimal layer tends to be.** What about situations that are more intricate, occurring between the above extreme cases? To explore this, we consider the following setting for simulation. We let

$$L = 12, \hat{\phi}_j = 1, \forall j \leq 9, \qquad \hat{\phi}_j = 0.1, \forall j \geq 10, \qquad j^* = 9,$$
$$c_j = 0.4, \forall j \leq 8, \qquad c_j = 0.6, \forall j \leq 10.$$

Then, we vary $c_{j^*}$, the weight assigned to the downstream relevant feature from $0.4$ to $0.6$, and plot $r_l$ vs $l$ (i.e., sample complexity indicator vs depth) under each value of $c_{j^*}$ in Figure 5. We observe that, for $c_{j^*} = 0.45, 0.5, 0.55$, the best sample complexity is achieved by some intermediate layer. Additionally, the optimal layer (corresponding to the bottom of the U-shaped curve) becomes lower as the weight assigned to the relevant feature decreases, which indicates a larger mismatch between the pretraining and downstream tasks.

**Challenges in locating the optimal layer.** Even in this simplified scenario, we observe that the depth of the optimal layer is influenced by various factors, including the position of the downstream-relevant feature, the strength of features in the downstream task, and the weights assigned to features during pretraining. The last one is further a function of features, noise and augmentations for pretraining data. In practical settings, we acknowledge that more factors may come into play, such as the model architecture, which varies across layers. Therefore, determining the exact optimal layer for downstream tasks is very challenging and represents an intriguing and valuable avenue for future exploration. We believe the analytical framework established in this paper, capable of expressing downstream sample complexity in closed form through various elements in pretraining and downstream tasks, and explaining several observed phenomena (e.g., those depicted in Figure 3), can significantly aid advancing research in this direction.

## C   EXPERIMENTAL DETAILS

### C.1   SETUPS

**Synthetic Experiments, CL.** We train two-layer (symmetrized) ReLu Networks with momentum SGD, with momentum set to 0.9. We use the spectral CL loss. (1) In setting 1, we set $d = 5, p = 20$ and $\phi_1 = \cdots = \phi_5 = 1$, indicating that all five features have equal strength. We set $\alpha_1, \alpha_2, \alpha_3, \alpha_4, \alpha_5$ to $0, 0.25, 0.5, 0.75, 1$, respectively, meaning that features are disrupted by the augmentation to different extents. We let $\sigma = 0.01$ and set learning rate to 0.05. (2) In setting 2, we set $d = 9, p = 20$, $\phi_i = 3.2/2^i, \forall i$, and $\sigma = 0.1$. We use learning rate 0.01.

**Synthetic Experiments, SCL.** The data distribution is the same as described in Definition 4.1. We let other coordinates be randomly drawn from $\{-0.001, 0.001\}$. We let $d = 5, p = 20, \sigma = 0.1$, learning rate = 0.01.

**MNIST-on-CIFAR-10.** The main data generation process is outlined in Section 5.1. To provide further details, after processing the digits in each image, we apply standard data augmentations, RandomResizedCrop and ColourDistortion. Additionally, we resize the digits to set their height to 16 pixels. We train ResNet-18 models. By default, we use a temperature of 0.5 and minimize the SimCLR loss with the Adam optimizer. Our training batch size is 512, with a learning rate of 0.001 and weight decay set to $1 \times 10^{-6}$. We train for 400 epochs. The projection head is a one-hidden layer MLP with an output dimension of 128, and the hidden dimension is set to match the output dimension of the ResNet-18 encoder.

**Coarse-to-fine transfer on CIFAR-100.** On CIFAR-100, we refer to the 10 super-classes as 'coarse' and the 100 classes as 'fine'. (1) SCL. The pretraining is conducted with the 10 coarse-grained labels, using the SCL loss in Khosla et al. (2020) with temperature 0.5. We use train a ResNet-18 with momentum SGD, using learning rate = 0.1, momentum = 0.9 and weight decay 1e-6. We train with batch size set to 512 for 400 epochs. The projection head is a one-hidden layer MLP with an output dimension of 128, and the hidden dimension is set to match the output dimension of the ResNet-18 encoder. (2) SL. We use momentum SGD, with learning rate = 0.1 and momentum = 0.9. We train for 200 epochs with batchsize 128 and weight decay 5e-4.

**Few-shot adaption on UrbanCars.** UrbanCars is constructed by Li et al. (2023), features multiple spurious correlations. The task is classifying images as either urban cars or country cars. Each image

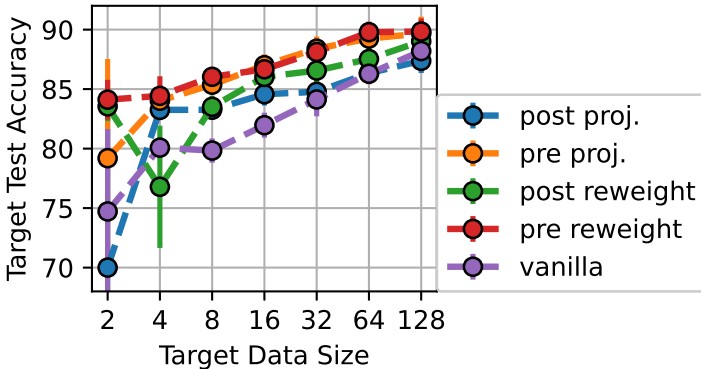

Figure 6: Target test accuracy with respect to the number of target examples for adaptation, comparing different representations.

has one background (BG) and one co-occurring object (CoObj). The BG is selected from either urban or country backgrounds, and the CoObj is selected from either urban or country objects. In the source distribution, for each class, images with common BG and CoObj constitute 90.25%, images with uncommon BG and common CoObj, or with common BG and uncommon CoObj, constitute 4.75%, and images with both uncommon BG and CoObj constitute 0.25%. This dataset presents a challenge due to multiple spurious correlations/shortcuts. We let the target distribution contains only the subpopulations that are most underrepresented in the source distribution, i.e., images with both uncommon BG and CoObj. We train the model on the source data using the SGD optimizer with a batch size of 128, a learning rate of 0.01, and weight decay of 0.000001 for 50 epochs. The linear layer is then trained on top of representations using a few (2 to 128) data from the target distribution. Following Chen et al. (2023), for the training of the linear layer, we employ the Adam optimizer Kingma & Ba (2014) with a batch size of 64 and train for 100 epochs. We tune both the learning rate and weight decay in the range of 0.1, 0.01, 0.001, and report the configuration that yields the best result. Each experiment is repeated 10 times, and we report the average. We use the implementation from Joshi et al. (2023) for these experiments.

**Experiments with reweighting heads.** All setups remain the same as before. We set the values of $\kappa$ to 1.05, 1.5, 1.2, and 1.01 for the four experiments in Table 2, from top to bottom, respectively.

### C.2   DETAILS RESULTS FOR THE LAST ROW OF TABLE 2

Figure 6 presents the target test accuracy with respect to the number of target examples for adaptation, comparing different representations. We observe that for both the projection head and the reweighting head, the representations before the head are better than those after, and also better than the vanilla approach, which does not add any head during pretraining. Furthermore, pre-reweighting-head representations outperform pre-projection-head representations, underscoring the potential of the reweighting head.

## D   ADDITIONAL EXPERIMENTS

### D.1   TWO EXPERIMENTS ABOUT FEATURE STRENGTH IN NATURAL IMAGES

**Dataset and Downstream Task.** The experiment is based on images in CIFAR-10, aiming to validate the second and third points outlined in Corollary 3.7 using real data. Given the subjective nature of defining features in natural images, we select the feature 'color,' which offers a straightforward and less controversial definition. This choice enables a controlled experiment with minimal modification to the original natural images. Our downstream task involves predicting whether a given image is categorized as 'red,' 'green,' or 'blue' based on the channel with the largest mean value. To vary the the strength of the color feature in the pretraining data, we consider the following two approaches.

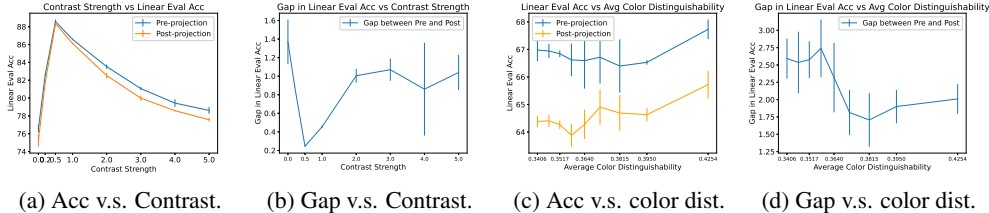

(a) Acc v.s. Contrast.     (b) Gap v.s. Contrast.     (c) Acc v.s. color dist.     (d) Gap v.s. color dist.

Figure 7: Accuracy and gap when the level of color distinguishability in the pretraining data is varied. (a)(c) are for approach 1, (b)(d) are for approach 2.

We note that the second one involves no change to the images themselves, making sure all of them are natural.

**1. Varying Color Distinguishability by Processing the Image**. During the training, we process each image as follows: Let $R$, $G$, and $B$ denote the average pixel value for the three channels. We first find the largest two values among $R$, $G$, and $B$ and calculates $u$ as their mean. For the matrix at each channel, denoted $M_R$, $M_G$, and $M_B$ for the red, green, and blue channels respectively, we update each matrix as follows:

$$M_R \leftarrow M_R \times \left( \frac{(R-u) \cdot \alpha + u}{R} \right),$$

$$M_G \leftarrow M_G \times \left( \frac{(G-u) \cdot \alpha + u}{G} \right),$$

$$M_B \leftarrow M_B \times \left( \frac{(B-u) \cdot \alpha + u}{B} \right).$$

Then, the values are capped between 0 and 1. $\alpha \geq 0$ is a parameter that we denoted as contrast strength. A larger $\alpha$ increases the values in the dominant channel while decreasing the values in other channels. Intuitively, larger $\alpha$ means a larger gap between the dominant channel and the other two channels, making the information about the dominant channel more obvious. We varied the $\alpha$ in the range of [0, 0.2, 0.5, 1, 2, 3, 4, 5].

**2. Varying Color Distinguishability by Selecting a Subset**. For each image, we choose the channel that has the greatest mean pixel value as the dominant channel, and compute the color distinguishability $D =$ (dominant channel's mean value / sum of the mean values for the three channels). We sort images in each class of CIFAR-10 based on $D$ from highest to lowest. Then we select 1000 images from each class starting from the K-th image to form the training set. Intuitively, a larger $K$ leads to images with higher average color distinguishability $D$ values being selected in the pretraining data, and a higher color distinguishability means that the information about the dominant channel is more obvious. We varied K in the range of [0, 500, 1000, 1500, 2000, 2500, 3000, 3500, 4000], which corresponds to the average color distinguishability of [0.4254, 0.3950, 0.3815, 0.3717, 0.3640, 0.3575, 0.3517, 0.3463, 0.3406].

**Training Details.** We train ResNet18 with a one hidden layer projection head with hidden dimension 2048 using SimCLR (Chen et al., 2020) on the selected trainset images for 400 epochs. We use Adam optimizer with learning rate 0.001, weight decay $10^{-6}$, and batch size 512. We used temperature 0.5.

**Results.** Figure 7 presents the results for the above two experiments. In both cases, we see that the gap between pre-projection and post-projection roughly show an decreasing-increasing trend. In other words, using pre-projection is more beneficial when either the color distringuishability is very high or very low, confirming our theoretical results in Corollary 3.7.

### D.2 Effect of Early Stopping

We examine how the linear evaluation accuracy changes during training on MNIST-on–CIFAR-10. The setting is consistent with the setting described in Section 5.1. Intuitively, early stopping should result in a model that is less specialized towards the training objective, potentially benefiting the

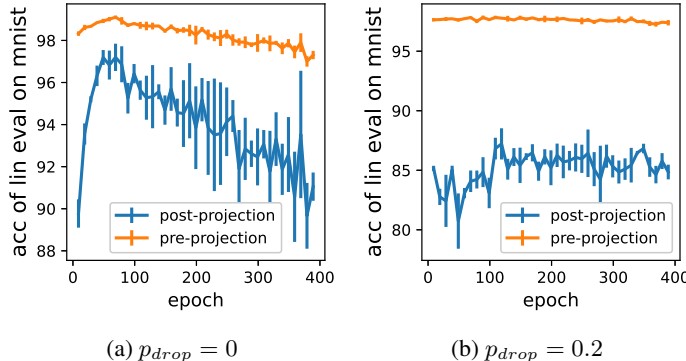

(a) $p_{drop} = 0$

(b) $p_{drop} = 0.2$

Figure 8: Linear evaluation accuracy during training. We see that early stopping reduces the gap between pre-projection and post-projection when $p_{drop} = 0$, but not when $p_{drop} = 0.2$.

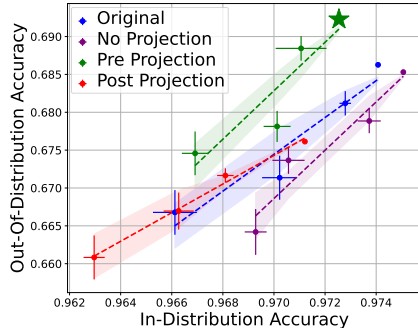

Figure 9: Pre-projection representations exhibit better robustness. OOD accuracy is evaluated across four shifted ImageNets.

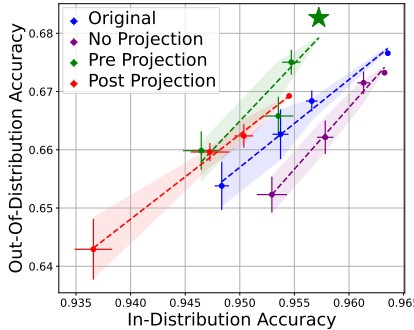

Figure 10: OOD-ID accuracy relation for models trained on Mixed-Rand.

downstream task when there is a misalignment. However, perhaps counter intuitively, the results in Figure 8 reveal that early stopping improves post-projection with good augmentation ($p_{drop} = 0$), but not with bad augmentation ($p_{drop} = 0.2$). A deeper analysis of training dynamics during the course of training is required to fully grasp the effect of early stopping.

### D.3 ROBUSTNESS TO DISTRIBUTION SHIFT ON IMAGENET

It is well-known that deep learning models trained on ImageNet Deng et al. (2009) tend to rely heavily on backgrounds rather than the main objects for making predictions (Xiao et al., 2020; Zhu et al., 2016), resulting in poor performance on test sets where backgrounds are random or absent.

Recently, Kirichenko et al. (2022) demonstrate that despite this poor performance, the representations before the linear classifier actually contain information about the objects. Consequently, training a new classifier on these representations using data that does not exhibit a correlation between backgrounds and classes can lead to improved out-of-distribution (OOD) accuracy. We demonstrate

that the quality of representations can be further enhanced by leveraging a non-linear projection head. We take the ImageNet pretrained ResNet-50 model, and fine-tune the model with an dditional projection head on ImageNet for 50 epochs. We find that the pre-projection representations result in an improved in-distribution vs. OOD accuracy relationship (which is a standard measurement of robustness (Taori et al., 2020)), as shown in Figure 9 where we compare it with post-projection representations, representations provided by the original pretrained model and representations obtained by fine-tuning the model for 50 epochs without the projection head.

**Experimental details.** Following Kirichenko et al. (2022), we use the datasets in Backgrounds Challenge based on the ImageNet-9 dataset (Xiao et al., 2020) along with the ImageNet-R (Hendrycks et al., 2021a) dataset. We use ImageNet-A (Hendrycks et al., 2021b) instead of Paintings-BG (Kirichenko et al., 2022) due to limited dataset availability. We consider Original, Mix-Rand, and FG-only datasets for Backgrounds Challenge. We finetune the ImageNet-pretrained ResNet-50 encoder along with an additional randomly initialized projection head and a randomly initialized linear classifier. The projection head is an MLP with one hidden layer of size 2048. We finetune the model for 50 epochs with batch size 256, momentum 0.9, weight decay $10^{-4}$, learning rate 0.01, and a learning rate scheduler with step size 30 and $\gamma = 0.1$. We also finetune the original ImageNet-pretrained ResNet-50 model with the same hyperparameters. During the training, we follow the same procedure as Kirichenko et al. (2022), using two train sets: Mixed-Rand and Combination of Mixed-Rand and Original. The sizes of Original and Mixed-Rand are the same. We use the same data preprocessing step as in Kirichenko et al. (2022). We train DFR on varied-sized random subsets of the training data with Mixed-Rand data sizes in the range of {5000, 10000, 20000, 45405} for 1000 epochs using SGD with full batch, learning rate 1, and weight decay equal to 100 / size of the data. For evaluation, we use four out-of-distribution datasets, Mixed-Rand, FG-Only, ImageNet-R, ImageNet-A, and one in-distribution dataset, Original. We average the accuracy of the four in-distribution datasets and present the out-of-distribution accuracy vs. in-distribution accuracy plot for the four settings: pre-projection, post-projection, original (representing original ImageNet-pretrained ResNet-50 model) and no-projection (representing finetuned ResNet-50 model without projection head). The results for Combination of Mixed-Rand and Original and Mixed-Rand are shown in Figures 9 and 10, respectively.

