# OpenReview forum: "Investigating the Benefits of Projection Head for Representation Learning"
_ICLR.cc/2024/Conference — ICLR 2024 poster_

### Official Review · Reviewer_p748 · 2023-10-27

**Soundness:** 4 excellent
**Presentation:** 3 good
**Contribution:** 3 good
**Rating:** 6
**Confidence:** 4

**Summary:**

This paper studied a very interesting question: what makes pre-projection representations better if they are not directly optimized? Based on theoretical analysis on some toy models, they proposed that the implicit bias of training algorithms makes deeper features more unequal, and hence lower layers tend to have more normalized and less specialized representations. Then they showed that lower layers are better in the following cases: (1) data augmentation disrupts useful feature; (2) downstream-relevant features are too weak/strong in pre-training. They also showed how this mechanism makes lower representations better for supervised contrastive learning and supervised learning. Finally, they conducted some experiments to verify their theoretical analyses.

**Strengths:**

1. This paper studied a very interesting question: what makes pre-projection representations better if they are not directly optimized?
2. Based on theoretical analysis on some toy models, they proposed that the implicit bias of training algorithms makes deeper features more unequal, and hence lower layers tend to have more normalized and less specialized representations.
3. Then they showed that lower layers are better in the following cases: (1) data augmentation disrupts useful feature; (2) downstream-relevant features are too weak/strong in pre-training.
4. They also showed how this mechanism makes lower representations better for supervised contrastive learning and supervised learning.

**Weaknesses:**

My main concern is that their data and model are too simple. But if similar models are commonly used, it may be okay.

**Questions:**

1. I want to know whether their data model (Def 3.1) is commonly used in literature? Even though the authors' theoretical analysis are insightful, I am worried that this data model is too simply to be applied to practical scenarios.
2. Most of the cases where lower representations are better arises from the inappropriate data augmentation (like Thm 4.2), namely the pre-training signal does not align with the downstream problem. In such cases, overfiting the pre-training data (what post-projection layers intend to do) may lead poor downstream performance. In addition to using pre-projection layers, can other regularization methods (weight decay, dropout, early stopping, etc.) also lead to satisfied performance even without projection head? I noticed that the author mentioned in the end of Sec 3 that the advantage of pre-projection representations diminishes when using weight decay.
3. The author mainly focused on two-layer neural networks. When there are multiple layers, how do the depth of representations affect the result? Is there any trade-off, like deeper layers have more representation power while less diversity/robustness? How to choose the appropriate depth of projection heads?

Overall, I think this is an insightful work and I am glad to raise my score if the authors can clarify their over-simple model and give a slightly deeper answers to the above questions.

---

> ### Author Response · Authors · 2023-11-23
> **Response to Reviewer p748 -- Part 1**
>
> > ## Relevance of our data model
>
>
> (1) Although simple, our model encompasses a wide array of aspects, demonstrating rich results. The analytical expression derived for the sample complexity indicator reveals intricate interactions among pretraining and downstream features, data augmentation, and noise.
>
> (2) Similar data models have been utilized in numerous studies, particularly within literature exploring feature learning. The data distribution defined in 3.1 is on a hyperrectangle, akin to a simpler model employed in (Saunshi 2021) that revealed compelling insights about contrastive learning. Additionally, when coupled with the data augmentation detailed in Definition 3.2, our data distribution effectively introduces noise into the input. A special case of our distribution is the sparse signal plus dense noise setting, extensively used in theoretical studies, with variations shown in Allen-Zhu 2020, Zou 2021, Shen 2022, Chen 2023 and more. This model naturally aligns with numerous machine learning scenarios and mirrors data structures in both image and language tasks (e.g., the second paragraph in Section 2.1 of Wen 2021 gives a brief overview of the sparse coding model's relevance). Moreover, Definition 3.2 also accounts for how augmentation can discourage the learning of a feature by decorrelating it within a positive pair. Our data distribution, combined with this augmentation modeling, is akin to models used in (Wen 2021), (Liu 2021), and (Xue 2023) for investigating feature learning in self-supervised learning.
>
>
> > ## “Most of the cases … also lead to satisfied performance even without projection head?.”
>
> We briefly discuss a few points: (1) weight decay. Our experiments suggest weight decay has a more negative than positive impact. Theoretically, as outlined in Section 3, weight decay encourages the model to be more ‘concise’, leading to the elimination of 'unnecessary' weights according to the training objective, thus removing additional features learned in pre-projection. Empirically, as verified in Fig 3d, increased weight decay worsens performance with inappropriate data augmentation and negates the advantages of pre-projection. (2) early stopping. Intuitively, early stopping should result in a model that is less specialized towards the training objective, potentially benefiting the downstream task when there is a misalignment. However, perhaps counter intuitively, our MNIST-on-CIFAR-10 experiments reveal that early stopping improves post-projection with good augmentation ($p_{drop}=0$), but not with bad augmentation ($p_{drop}=0.2$).The corresponding plots are included in Fig 8, where we plot the linear evaluation accuracy against number of training epochs. A deeper analysis of training dynamics during the course of training is required to fully grasp the effect of early stopping. (3) The impact of dropout might not be immediately evident, but we believe that future research could utilize the analytical framework presented in our paper to investigate the effects of dropout.

---

> ### Author Response · Authors · 2023-11-23
> **Response to Reviewer p748 -- Part 2**
>
> > ## “The author mainly focused on two-layer neural networks…”
>
> This is a great question, but providing a clear answer for deep non-linear models poses a challenge with our current understanding of deep neural networks in the community. Nonetheless, we can offer some insights through deep linear models. As mentioned in section 4.1, Theorem 3.4 can be expanded to multiple-layer models with any loss function.
>
> (1) **Similar conclusions to those in Section 3.1 can be drawn**. For a $L$-layer network, we have $W_l(t)^\top W_l(t) = W_{l+1}(t) W_{l+1}(t)^\top$. Using this, one can further derive the sample complexity indicator for the setting considered in Section 3 as follows
>
> \begin{align}
>     r_l=\frac{ \sum_{j=1}^p c_j^{2l/L} \hat{\phi_j}^2 }{ c_{j^*}^{2l/L}\hat{\phi}_{j^*}^2 }
> \end{align}
>
> where
>
> \begin{align}
>     c_j = \frac{(1-\alpha_j)\phi_j}{\phi_j^2 + \sigma^2} ~~~ \text{if}~~~ j\in  \{j_1, \dots, j_{\min\{d, p\}}\} ~~~\text{else}~~~ 0
> \end{align}
>
> The definitions of $j_1, \dots, j_{\min\{d, p\}}$ remain the same as in Theorem 3.5, and other quantities are consistent with the definitions provided in Section 3. Here, $c_j$ represents the weight the full model would allocate to the j-th feature. Although the expression appears complex, some intuition can be gleaned from extreme scenarios: If $c_{j^*}$ is the largest, indicating that the full model assigns the most weight to the downstream relevant feature, $r_l$ decreases with $l$. This means one should just use the final-layer representations. Conversely, if $c_{j^*}$ is the smallest among non-zero $c_j$'s, indicating that the full model assigns the least weight to the downstream relevant feature, $r_l$ increases with $l$. In this case, using the lowest layer would be preferable. Applying these observations in conjunction with the relationship between $c_j$ and $\alpha_j, \phi_j$, similar conclusions to those in Corollary 3.7 regarding the impact of augmentation and feature strength for multi-layer models can be drawn.
>
> (2) **The greater the mismatch between the pretraining and downstream tasks, the lower the optimal layer tends to be**. What about situations that are more intricate, occurring between the above extreme cases? By setting specific values for $c_j$ and $\hat{\phi}_j$’s, and plotting the sample complexity indicator $r_l$ against the depth $l$, we observe instances where intermediate layers perform better (which we believe are the most common cases in practice). Please refer to Fig 6, where we see a U-shaped curve. Additionally, we conduct further exploration by fixing the weights for all other features and vary the weight for the downstream relevant features (the exact configurations are detailed in Appendix C). Fig 6 shows that the optimal layer (corresponding to the bottom of the U-shaped curve) becomes lower as the weight assigned to the relevant feature decreases, which indicates a larger mismatch between the pretraining and downstream tasks.
>
> (3) **Challenges in locating the optimal layer**: Even in this simplified scenario, we observe that the depth of the optimal layer is influenced by various factors, including the position of the downstream-relevant feature, the strength of features in the downstream task, and the weights assigned to features during pretraining. The last one is further a function of features, noise and augmentations for pretraining data. In practical settings, we acknowledge that more factors may come into play, such as the model architecture, which varies across layers. Therefore, determining the exact optimal layer for downstream tasks is very challenging and represents an intriguing and valuable avenue for future exploration. **We believe the analytical framework established in this paper, capable of expressing downstream sample complexity in closed form through various elements in pretraining and downstream tasks, and explaining several observed phenomena (e.g., those depicted in Figure 3), can significantly aid advancing research in this direction**.

---

> ### Author Response · Authors · 2023-11-23
> **Reference**
>
> Saunshi, Nikunj, et al. "Understanding contrastive learning requires incorporating inductive biases." International Conference on Machine Learning. PMLR, 2022.
>
> Shen, Ruoqi, Sébastien Bubeck, and Suriya Gunasekar. "Data augmentation as feature manipulation." International conference on machine learning. PMLR, 2022.
>
> Xue, Yihao, et al. "Which Features are Learnt by Contrastive Learning? On the Role of Simplicity Bias in Class Collapse and Feature Suppression." arXiv preprint arXiv:2305.16536 (2023).
>
> Chen, Jinghui, Yuan Cao, and Quanquan Gu. "Benign overfitting in adversarially robust linear classification." Uncertainty in Artificial Intelligence. PMLR, 2023.
>
> Zou, Difan, et al. "Understanding the generalization of adam in learning neural networks with proper regularization." arXiv preprint arXiv:2108.11371 (2021).
>
> Allen-Zhu, Zeyuan, and Yuanzhi Li. "Towards understanding ensemble, knowledge distillation and self-distillation in deep learning." arXiv preprint arXiv:2012.09816 (2020).
>
> Wen, Zixin, and Yuanzhi Li. "Toward understanding the feature learning process of self-supervised contrastive learning." International Conference on Machine Learning. PMLR, 2021.

---

### Official Review · Reviewer_B9NE · 2023-10-31

**Soundness:** 3 good
**Presentation:** 3 good
**Contribution:** 3 good
**Rating:** 6
**Confidence:** 3

**Summary:**

This paper investigates the effectiveness of the projection head in self supervised contrastive learning, supervised contrastive learning and supervised learning. It provides theoretical analysis of the quality and robustness of the learned representations and their generalizability in simple linear and nonlinear models. Theoretical results are supported with experimental evaluation on several image datasets.

**Strengths:**

- Thorough theoretical analysis and interesting insights.
- Extensive related work.

**Weaknesses:**

- Experimental evaluation is done on very simple networks and small datasets. Although the results nicely support the theoretical results, it might be beneficial to include one more complex experiment.
- The paper could benefit from a discussion on limitations and assumptions of the analysis.

**Questions:**

- In Definition 3, could you add details about $\phi_i$?
- Theorem 3.6 essentially tells us when it is beneficial to use pre and post projection representations, with concrete guidelines given in Corollary 3.7. Do these results hold for non linear models as well? As a stretch question, I was wondering how could one infer the downstream-relevant features and the weights of features in practice? In other words, is there a way to use your results to identify what features will be relevant for the downstream task? I am aware this might be out of scope of this work.
- What is $\alpha_i$ in setting 2 in Sec 5.1? Could you comment on the results in Fig 1 right, where $\phi_i$ = 0.2 and 0.4, in particular, why do we see the spikes in weights in pre-features for $\phi_i$ = 0.2 and post-features for $\phi_i$ = 0.4?
- I do not understand the experiment in Fig 3b. In the text your say that “Figure 3b shows the downstream accuracy against $p_{drop}$ and s=1”. However, if s=1, it means that you only use MNIST images as input to the augmentation, and then you additionally drop the digit with probability $p_{drop}$. Doesn’t that mean your augmented image is completely black if $p_{drop}=1$?
- In Table 2, WaterBird SL results are almost the same, especially for no projection and pre-projection. Why is that?
- Could you explain better what exactly is plotted in Figure 4?

---

> ### Author Response · Authors · 2023-11-23
>
> > ## “Experimental evaluation is done on very simple networks and small datasets. Although the results nicely support the theoretical results, it might be beneficial to include one more complex experiment.”
>
> (1) Many empirical studies have already demonstrated the effects of the projection head and data augmentation via large scale experiments. For instance, Chen et al. (2020) in Table 3, Bordes et al. (2023) in Figure 2, and Rashtchian et al. (2023) in Table 2 present experimental results for self-supervised learning, showing that: a) Information suppressed by data augmentation, such as Color, Hue, Rotation, and Spot, is more retrievable in pre-projection representations. b) Consequently, if the downstream task requires this suppressed information, utilizing pre-projection is preferable. They also provide results about supervised learning. Therefore, our objective is not to replicate these results but to provide theoretical understanding for them.
>
> (2) Additionally, we conducted another set of experiments on CIFAR-10, maintaining the same scale as our MNIST-on-CIFAR-10 experiment in the original version, but incorporating more natural methods to control the feature's strength. The experiment details can be found in Appendix D.1. In summary, we established the downstream task as categorizing CIFAR-10 images as more ‘red,’ ‘green,’ or ‘blue.’ To control the color distinguishability (the clarity in identifying a color in images), we utilized two approaches: 1) manipulating the images, or 2) selecting subsets with varying color distinguishability. This effectively controls the downstream-relevant feature’s strength. Our observations in Fig 7(b)(c) revealed that, in both scenarios, the advantage of pre-projection is more pronounced when the color feature is either too weak or too strong.
>
> (3) Moreover, our last experiment in section 5 focuses on ImageNet, demonstrating how one can enhance the utility of ImageNet-pretrained model’s representations under distribution shifts. This is achieved by finetuning the model with an additional projection head and subsequently removing the projection head. We believe this finding is highly practical, offering a simple and efficient way to improve the model's robustness on large datasets.
>
>
> “The paper could benefit from a discussion on limitations and assumptions of the analysis.”
>
> > ## Discussion on limitations and assumptions of the analysis
>
> One limitation arises from the assumptions made in the analysis of non-linear models, where we simplified the analysis by considering a diagonalized network rather than a fully connected one. This difference may affect training dynamics, posing challenges in characterizing the learning of additional features in a fully connected multi-layer non-linear network. Nevertheless, we maintain that insights drawn from our proof in Appendix A.4 (that ReLU function’s threshold behavior filters gradients to preserve additional features in the pre-projection layer) should still apply and could assist in analyzing this more complex scenario.
> Moreover, while the experiment in Fig 4 demonstrates the benefits of the projection head amid distribution shifts, our discussion in Section 4.1 provides a general insight based on linear models, rather than characterizing this effect for nonlinear modes. The technical challenge arises from our analysis of the diagonalized nonlinear model, which relies on the coordinate-wise symmetric property of the input distribution. To study distribution shift, one often needs to consider data distributions with spurious correlations, (e.g., the data model in Sagawa 2020), where this property no longer holds due to correlations between features. Further exploration in this direction would be valuable to deepen the theoretical understanding of the observations presented in Section 4.1.
>
> > ## $\phi_i$ in Definition 3
>
> The explanation of the meaning of $\phi_i$ can be found in the paragraph following Definition 3.2. In this data model, each coordinate represents a feature, and the corresponding $\phi_i$ represents the magnitude (strength) of that feature.

---

> ### Author Response · Authors · 2023-11-23
>
> > ## “Theorem 3.6 essentially tells us when it is beneficial to use pre and post projection representations, with concrete guidelines given in Corollary 3.7. Do these results hold for non linear models as well?”
>
>
> Empirically, as depicted in Fig 1, these findings also hold true for non-linear models. The model employed for Fig 1 is a two-layer fully connected neural network. In Fig 1 Left, where all features have the same strength but are disrupted differently by augmentation, we see that pre-projection treats features more equally compared to post-projection, and larger disruptions from augmentation result in more significant differences between pre-projection and post-projection weights. These would lead to the first conclusion of Corollary 3.7. In Fig 1 Right, the feature-moderating effect demonstrated aligns with Theorem 3.5, showcasing that both the strongest and weakest features are weighted the least. This observation essentially supports the second and third conclusions of Corollary 3.7. Overall, our findings in Corollary 3.7 apply to non-linear models. However, explicitly and theoretically characterizing these phenomena in non linear models remains a challenging aspect, which points toward future directions.
>
>
> > ## “As a stretch question, I was wondering how could one infer the downstream-relevant features and the weights of features in practice? In other words, is there a way to use your results to identify what features will be relevant for the downstream task? I am aware this might be out of scope of this work.”
>
> Although beyond the scope of this paper, identifying relevant features and quantifying their assigned weights could potentially be achieved using existing analysis tools, such as visual explanation tools (Selvaraju 2016, which visualizes the regions of an image that contribute most to a neural network's decision-making) or PCA-based analysis (Jiang 2023, which defines features of an image to be the projection onto the principal components of different models’ last-layer activations). In future research, if these methods can be applied and combined with the findings of our paper, they might enable us to adjust the weighting of features to replicate the effects of a projection head, essentially reducing the reliance on using a projection head.
>
>
> > ## “What is alpha_i  in setting 2 in Sec 5.1?”
>
> $\alpha_i$ in setting 2 in Sec 5.1 is consistent with Definition 3.2 for data augmentation. For each $i$, the data augmentation randomizes the $i$-th feature with a probability of $\alpha_i$. Therefore, larger $\alpha_i$ means the data augmentation disrupts this feature more, consequently disencouraging the learning of this feature more. In setting 2, we set all $\alpha_i$’s to the same, meaning that the augmentation treats all features equally. This enables us to isolate and study the effect of feature strength.
>
> > ## “Could you comment on the results in Fig 1 right, where phi_i = 0.2 and 0.4, in particular, why do we see the spikes in weights in pre-features for phi_i = 0.2 and post-features for phi_i = 0.4?”
>
> Fig 1 Right aims to demonstrate how the model moderates the features as captured in Theorem 3.5, where both the strongest and weakest features are weighted the least, while features with intermediate strengths are weighted the most. This effect is precisely depicted in the figure. The spikes observed at intermediate strengths (e.g., 0.2 and 4) signify that these features are weighted more than those that are either stronger or weaker than them, aligning exactly with our expectations based on Theorem 3.5. However, using a two-layer ReLU network may result in some differences compared to a linear model. For instance, the positions of spikes in pre-projection and post-projection do not align, unlike in a linear model. Understanding these differences might require a more detailed analysis of nonlinear models for future research. Nonetheless, our primary goal of validating the overall behavior is achieved.

---

> ### Author Response · Authors · 2023-11-23
>
> > ## Setting of MNIST-on-CIFAR-10
>
> The original text was not entirely accurate, and we apologize for any confusion caused. However, the sample images presented in Fig 3 (a) accurately reflect our process: We only perform a weighted sum between CIFAR-10 and MNIST on the digit's body area, while preserving the other parts of the image unchanged from the CIFAR-10 image. Essentially, we modify the transparency of the digit's body. The code used to blend these two images is roughly as follows
>
> ```
> new_image = (mnist_image <= 0) * cifar_image +  (mnist_image > 0) * (s * mnist_image + (1-s)* cifar_image)
> ```
> Here after being converted to RGB using .convert('RGB') the MNIST image holds negative values for areas that are not part of the digit’s body. Therefore, varying s does not change the parts of the CIFAR-10 image that do not overlay with the digit’s body.
>
> We revised the experiment description for this part in Section 5.1, highlighted in blue.
>
> > ## “In Table 2, WaterBird SL results are almost the same, especially for no projection and pre-projection. Why is that?”
>
> This interesting phenomenon might be explained as follows. SCL loss directly operates on the post-projection representations, explicitly defining their expected arrangement. In contrast, SL loss is computed on the prediction (logits), one layer after the representations, and therefore does not explicitly define the arrangement of the representations. As a result, SL is ‘milder’ than SCL, allowing the representations to be less specialized towards the training task (this is reflected in SL's higher worst-group accuracy in Table 2). This, along with the relatively simple nature of the WaterBirds dataset, allows SL’s post-projection representations to already learn most features necessary for the downstream task, making pre-projection’s advantage less obvious.
>
> > ## “Could you explain better what exactly is plotted in Figure 4?”
>
> The plot in Figure 4 represents the OOD-vs-ID evaluation typically employed in studies on distribution shift robustness (e.g., see Taori 2020, Miller 2021, Radford 2021). We specifically compared the robustness of representations learned by different methods on ImageNet. Following Kirichenko 2022, we consider 4 different OOD benchmarks: Mixed-Rand, FG-Only, ImageNet-R, and ImageNet-A, and let the OOD accuracy be the average over the four datasets. As Mixed-Rand and FG-Only contain only nine ImageNet classes, we define ID accuracy as the accuracy on the corresponding nine classes in the original ImageNet (ImageNet-9) for comparison.
>
> Our aim is to assess how robust representations are against distribution shifts in different models trained on the original ImageNet. To achieve this, we followed Kirichenko 2022 which shows that training a linear classifier on the representations, using a mix of data from original ImageNet-9 and Mixed-Rand can lead to decent accuracy across OOD datasets. Therefore we use the OOD accuracy of this linear classifier to indicate the quality of the representations. Additionally, as suggested by Taori 2020, we also take into account the ID accuracy by plotting the OOD-ID relation to have a more comprehensive comparison. To do this, we varied the sample size for training the linear classifier to obtain different OOD-ID pairs and plotted them to observe the relationship. Our findings reveal that pre-projection (where the publicly available ImageNet pretrained model is fine-tuned on ImageNet for 50 epochs with an additional projection head which subsequently removed) not only achieves the highest overall OOD accuracy (the star in Figure 4) but also demonstrates a superior OOD-ID relationship. Specifically, (1) under the same ID distribution, it achieves better OOD distribution, and (2) compared to others, its OOD-ID slope is larger, indicating that a greater gain in OOD accuracy can be achieved by improving ID accuracy.

---

> ### Author Response · Authors · 2023-11-23
> **Reference**
>
> Miller, John P., et al. "Accuracy on the line: on the strong correlation between out-of-distribution and in-distribution generalization." International Conference on Machine Learning. PMLR, 2021.
>
> Radford, Alec, et al. "Learning transferable visual models from natural language supervision." International conference on machine learning. PMLR, 2021.
>
> Taori, Rohan, et al. "Measuring robustness to natural distribution shifts in image classification." Advances in Neural Information Processing Systems 33 (2020): 18583-18599.

---

### Official Review · Reviewer_EBwM · 2023-11-02

**Soundness:** 3 good
**Presentation:** 3 good
**Contribution:** 2 fair
**Rating:** 5
**Confidence:** 4

**Summary:**

The paper delves into a nuanced aspect of neural network architecture design, specifically the use of a projection head during the training phase. This technique has garnered attention due to its empirical success in enhancing representation quality. The core methodology involves appending a projection head atop the encoder during training, which is subsequently discarded, favoring the pre-projection layer representations for inference tasks.

Despite its proven practical effectiveness, a comprehensive theoretical understanding of why the projection head enhances representation learning remains underdeveloped. The paper aims to bridge this gap by dissecting the mechanics of the projection head and elucidating its impact on the learning dynamics of neural networks. This investigation is critical as it addresses a disconnect between empirical practices and their theoretical foundations in the field of deep learning.

The projection head's primary role is hypothesized to act as a regularization mechanism, potentially aiding in learning more generalizable and robust features. By expanding the representational capacity during training, the projection head could encourage the encoder to learn a broader set of features, some of which may be discarded during the projection phase but still contribute to a richer feature space in the pre-projection layer. Moreover, the projection head could serve to disentangle the feature space, making it easier for the network to differentiate between relevant and irrelevant features. This disentanglement might facilitate better generalization to new, unseen data by reducing overfitting to the idiosyncrasies present in the training dataset.

**Strengths:**

**Empirical and Theoretical Integration**:
The paper bridges the gap between empirical success and theoretical understanding by critically investigating the role of the projection head in representation learning. By scrutinizing a technique that has demonstrated practical effectiveness without a solid theoretical foundation, the paper contributes to a more profound understanding of neural network architectures, potentially guiding future designs with a better-informed rationale.

**Regularization and Feature Representation**:
It hypothesizes that the projection head is a regularization mechanism, allowing the encoder to explore a wider feature space during training. This could lead to more robust and generalizable representations, as the encoder is encouraged to capture a broader and more nuanced feature landscape. The paper's exploration of this aspect could elucidate how neural networks can be trained more effectively to learn generalizable features.

**Feature Disentanglement**:
The projection head's potential to disentangle the feature space is a significant strength of the paper's hypothesis. By facilitating a clearer separation of relevant and irrelevant features, the projection head might aid in reducing overfitting and improving the model's ability to generalize to unseen data. This aspect of the paper could contribute valuable insights into how neural networks can be made more interpretable and reliable.

**Weaknesses:**

**Potential Overfitting Risks**:
The introduction of a projection head could potentially lead to overfitting, especially if not properly regularized or if used in conjunction with datasets that have a high degree of noise or variability. The paper should address these risks and propose strategies to mitigate them.

**Generalizability and Applicability**:
The projection head's effectiveness might vary across different architectures, tasks, and data modalities. The paper could benefit from a more detailed exploration of these variations to understand where the projection head is most beneficial and where it might be detrimental. This would enhance the paper's technical depth and practical applicability.

**Questions:**

1. **How does the architecture of the projection head influence the learning dynamics and final representation quality?**
   - The design choices within the projection head (e.g., the number of layers, types of activations, dropout rates) likely have a profound impact on its efficacy as a regularizer and feature disentangler. What are the optimal architectural configurations for different types of data and tasks? Investigating this could provide more nuanced guidelines for practitioners and lead to a deeper theoretical understanding of the projection head's role.

2. **What is the impact of the projection head on the interpretability of the learned representations?**
   - While the projection head might aid in learning more generalizable features, its impact on the interpretability of these features is unclear. Do the representations learned with a projection head offer better clarity in terms of feature importance or contribution to the final decision? Understanding this could bridge the gap between performance and explainability in neural networks.

3. **Can the benefits of the projection head be replicated or enhanced by alternative or complementary techniques?**
   - Are there other methods or architectural innovations that could either replicate the benefits of the projection head or enhance its effects? For instance, could certain types of normalization, attention mechanisms, or even different training paradigms offer similar or greater benefits regarding feature representation and generalization? Exploring this could lead to a broader set of tools for improving neural network training beyond the projection head.

Delving into these questions could significantly enhance the paper's contribution, offering both a deeper theoretical understanding and more practical guidelines for employing projection heads in neural network training.

---

> ### Author Response · Authors · 2023-11-23
> **Response to Reviewer EBwM -- Part 1**
>
> > ## “Potential Overfitting Risks … The paper should address these risks and propose strategies to mitigate them”
>
> (1) it's important to note that we're not proposing a method; rather, we're investigating the roles of the projection head. (2) We have shown that the projection head acts more as a form of regularization rather than exacerbating overfitting. It prevents the representations from being overly specialized to the pre-training task, which can potentially have some misalignment with the downstream task. Therefore, the paper precisely delves into studying how the projection head addresses these aspects mentioned by the reviewer.
>
>
> > ## “Generalizability and Applicability … where the projection head is most beneficial and where it might be detrimental.”
>
> (1) Regarding the 'effectiveness across different tasks' and 'where the projection head is most beneficial and where it might be detrimental,' our paper precisely focuses on these aspects. It delves deeply into how the benefits of the projection head are reliant on the relationship between training and downstream tasks. The paper covers various cases, including the effects of data augmentation and feature strength. Additionally, we explore different types of pre-training losses, encompassing both self-supervised CL, supervised CL, and standard supervised learning.
>
> (2) Regarding architectures and data modalities, it's important to acknowledge the vast scope of these topics, which makes comprehensive coverage in a single paper unfeasible, despite their significance. Our primary objective revolves around offering the first analytical framework to study and theoretically characterize the projection head's role and its interaction with features and data augmentations. We believe this contribution merits recognition.
>
>
> > ## ‘How does the architecture … projection head’s role’
>
> In the following two points, we discuss how our paper's analysis provides insight regarding the number of layers in the projection head and activation.

---

> ### Author Response · Authors · 2023-11-23
> **Response to Reviewer EBwM -- Part 2**
>
> > ## Effect of the number of layers
>
>
> To grasp the impact of the number of layers, we essentially aim to determine when it becomes beneficial to extract representations from layers preceding the penultimate layer, and, if so, how many layers prior to that point? Providing a clear answer for deep non-linear models poses a challenge with our current understanding of deep neural networks in the community. Nonetheless, we can offer some insights through deep linear models. As mentioned in section 4.1, Theorem 3.4 can be expanded to multiple-layer models with any loss function.
> (1) **Similar conclusions to those in Section 3.1 can be drawn**. For a $L$-layer network, we have $W_l(t)^\top W_l(t) = W_{l+1}(t) W_{l+1}(t)^\top$. Using this, one can further derive the sample complexity indicator for the setting considered in Section 3 as follows
>
> \begin{align}
>     r_l=\frac{ \sum_{j=1}^p c_j^{2l/L} \hat{\phi_j}^2 }{ c_{j^*}^{2l/L}\hat{\phi}_{j^*}^2 }
> \end{align}
>
> where
>
> \begin{align}
>     c_j = \frac{(1-\alpha_j)\phi_j}{\phi_j^2 + \sigma^2} ~~~ \text{if}~~~ j\in  \{j_1, \dots, j_{\min\{d, p\}}\} ~~~\text{else}~~~ 0
> \end{align}
>
> The definitions of $j_1, \dots, j_{\min\{d, p\}}$ remain the same as in Theorem 3.5, and other quantities are consistent with the definitions provided in Section 3. Here, $c_j$ represents the weight the full model would allocate to the j-th feature. Although the expression appears complex, some intuition can be gleaned from extreme scenarios: If $c_{j^*}$ is the largest, indicating that the full model assigns the most weight to the downstream relevant feature, $r_l$ decreases with $l$. This means one should just use the final-layer representations. Conversely, if $c_{j^*}$ is the smallest among non-zero $c_j$'s, indicating that the full model assigns the least weight to the downstream relevant feature, $r_l$ increases with $l$. In this case, using the lowest layer would be preferable. Applying these observations in conjunction with the relationship between $c_j$ and $\alpha_j, \phi_j$, similar conclusions to those in Corollary 3.7 regarding the impact of augmentation and feature strength for multi-layer models can be drawn.
>
> (2) **The greater the mismatch between the pretraining and downstream tasks, the lower the optimal layer tends to be**. What about situations that are more intricate, occurring between the above extreme cases? By setting specific values for $c_j$ and $\hat{\phi}_j$’s, and plotting the sample complexity indicator $r_l$ against the depth $l$, we observe instances where intermediate layers perform better (which we believe are the most common cases in practice). Please refer to Fig 6, where we see a U-shaped curve. Additionally, we conduct further exploration by fixing the weights for all other features and vary the weight for the downstream relevant features (the exact configurations are detailed in Appendix C). Fig 6 shows that the optimal layer (corresponding to the bottom of the U-shaped curve) becomes lower as the weight assigned to the relevant feature decreases, which indicates a larger mismatch between the pretraining and downstream tasks.
>
> (3) **Challenges in locating the optimal layer**: Even in this simplified scenario, we observe that the depth of the optimal layer is influenced by various factors, including the position of the downstream-relevant feature, the strength of features in the downstream task, and the weights assigned to features during pretraining. The last one is further a function of features, noise and augmentations for pretraining data. In practical settings, we acknowledge that more factors may come into play, such as the model architecture, which varies across layers. Therefore, determining the exact optimal layer for downstream tasks is very challenging and represents an intriguing and valuable avenue for future exploration. **We believe the analytical framework established in this paper, capable of expressing downstream sample complexity in closed form through various elements in pretraining and downstream tasks, and explaining several observed phenomena (e.g., those depicted in Figure 3), can significantly aid advancing research in this direction**.
>
> > ## Role of ReLu activation
>
> In both Thm 3.8 and 4.2, we have shown how ReLu activation enables the pre-projection layer to learn features that are not learned at all by the post-projection layer. This occurs in both self-supervised CL, supervised CL, and standard supervised learning, whereas linear models cannot exhibit this behavior. Drawing intuition from our proof in Appendix A.4, the ReLu function exhibits a threshold-like behavior, effectively filtering the gradient passed from post-projection to prevent the elimination of the additional feature in the pre-projection layer. It could be intriguing for future research to explore whether different types of nonlinearity (e.g., GeLu, Leaky ReLu) yield varying effects and potentially facilitate a more tailored design for the projection head.

---

> ### Author Response · Authors · 2023-11-23
> **Response to Reviewer EBwm -- Part 3**
>
> > ## impact of the projection head on the interpretability of the learned representations
>
> Interpretability isn't our primary focus in this paper. We do encourage future research to leverage our analytical framework for exploring interpretability, given that our framework offers an analytical expression of the sample complexity indicator for each layer, involving various aspects of features and data augmentation. Nevertheless, this topic falls outside the scope of our current study.
>
> In terms of feature importance and contribution, our analysis in Thm 3.5 shows that the features are represented more equally at pre-projection, and Thm 3.8 and Thm 4.2 show that pre-projection can learn more features. Consequently, when there is a misalignment between pretraining and downstream objectives, the feature important for the downstream task is more likely to 'survive' in pre-projection.
>
>
>
> > ##  “Can the benefits of the projection head be replicated …”
>
>
> Our study paves the way for exploring these topics. Generally, the projection head can be utilized as a black box or a more general technique to improve representations. Our findings precisely identify the reasons behind the benefits of the projection head, allowing for the implementation of more targeted techniques in various scenarios. These may include improved data augmentation strategies (to avoid disturbing the downstream relevant feature), larger representation dimensions (to learn more features), and employing early stopping techniques (to prevent the representations from being overly specialized towards the training task). We believe exploring alternative techniques is an interesting direction for future research, and our work serves the first step towards this direction by demystifying the role of the projection head.

---

### Official Review · Reviewer_Kpzq · 2023-11-04

**Soundness:** 3 good
**Presentation:** 2 fair
**Contribution:** 2 fair
**Rating:** 5
**Confidence:** 4

**Summary:**

This paper analyzed an important technique in contrastive learning: the projection head. The authors theoretically demonstrated the benefits of the projection layer via a simplified model. The theoretical analysis showed that lower layers represent features more evenly in linear networks and can represent more features in non-linear networks, which implies better generalization performance in downstream tasks. Empirically, they verify the theoretical findings on synthetic and real-world datasets.

**Strengths:**

1. The projection layer is one of the most important techniques in contrastive learning and the mechanism behind it is still under-explored. Consequently, this paper addresses an important problem.
2. The theoretical analysis in this paper looks solid and insightful. And the empirical results verify the theoretical findings.

**Weaknesses:**

1. The theoretical analysis in this paper is based on a two-layer model. Is it possible to extend the results to multiple-layer networks? For example, should we discard other layers except for the projection head in downstream tasks in the deep networks? It would be better to provide more discussions about that.
2. This paper demonstrates the benefits of the projection head. However, we can observe that the designs of the projector (e.g., the layers and the dimensions) also have a significant influence on the downstream performance. Is it possible to provide some insights about the design of the projector based on the theoretical analysis in this paper?
3. As stated in this paper, the pre-projection representations are preferred in three different scenarios and the findings are verified on the synthetic datasets. However, the authors do not show similar results (e.g., the influence of data augmentations) on the real-world datasets. It would be better to provide more empirical findings on real-world datasets.
4. I note that the abstract on the OpenReview website is different from that on the pdf file, which should be corrected.
5. The forms of references are inconsistent. For example, some of the conferences are full titles while others are abbreviations.

**Questions:**

see my comments above.

---

> ### Author Response · Authors · 2023-11-23
> **Response to Reviewer Kpzq -- Part 1**
>
> > ## Multi-layer
>
> This is a great question, but providing a clear answer for deep non-linear models poses a challenge with our current understanding of deep neural networks in the community. Nonetheless, we can offer some insights through deep linear models. As mentioned in section 4.1, Theorem 3.4 can be expanded to multiple-layer models with any loss function. The following discussion is also added to Appendix C.
>
> (1) **Similar conclusions to those in Section 3.1 can be drawn**. For a $L$-layer network, we have $W_l(t)^\top W_l(t) = W_{l+1}(t) W_{l+1}(t)^\top$. Using this, one can further derive the sample complexity indicator for the setting considered in Section 3 as follows
>
> \begin{align}
>     r_l=\frac{ \sum_{j=1}^p c_j^{2l/L} \hat{\phi_j}^2 }{ c_{j^*}^{2l/L}\hat{\phi}_{j^*}^2 }
> \end{align}
>
> where
>
> \begin{align}
>     c_j = \frac{(1-\alpha_j)\phi_j}{\phi_j^2 + \sigma^2} ~~~ \text{if}~~~ j\in  \{j_1, \dots, j_{\min\{d, p\}}\} ~~~\text{else}~~~ 0
> \end{align}
>
> The definitions of $j_1, \dots, j_{\min\{d, p\}}$ remain the same as in Theorem 3.5, and other quantities are consistent with the definitions provided in Section 3. Here, $c_j$ represents the weight the full model would allocate to the j-th feature. Although the expression appears complex, some intuition can be gleaned from extreme scenarios: If $c_{j^*}$ is the largest, indicating that the full model assigns the most weight to the downstream relevant feature, $r_l$ decreases with $l$. This means one should just use the final-layer representations. Conversely, if $c_{j^*}$ is the smallest among non-zero $c_j$'s, indicating that the full model assigns the least weight to the downstream relevant feature, $r_l$ increases with $l$. In this case, using the lowest layer would be preferable. Applying these observations in conjunction with the relationship between $c_j$ and $\alpha_j, \phi_j$, similar conclusions to those in Corollary 3.7 regarding the impact of augmentation and feature strength for multi-layer models can be drawn.
>
> (2) **The greater the mismatch between the pretraining and downstream tasks, the lower the optimal layer tends to be**. What about situations that are more intricate, occurring between the above extreme cases? By setting specific values for $c_j$ and $\hat{\phi}_j$’s, and plotting the sample complexity indicator $r_l$ against the depth $l$, we observe instances where intermediate layers perform better (which we believe are the most common cases in practice). Please refer to Fig 6, where we see a U-shaped curve. Additionally, we conduct further exploration by fixing the weights for all other features and vary the weight for the downstream relevant features (the exact configurations are detailed in Appendix C). Fig 6 shows that the optimal layer (corresponding to the bottom of the U-shaped curve) becomes lower as the weight assigned to the relevant feature decreases, which indicates a larger mismatch between the pretraining and downstream tasks.
>
> (3) **Challenges in locating the optimal layer**: Even in this simplified scenario, we observe that the depth of the optimal layer is influenced by various factors, including the position of the downstream-relevant feature, the strength of features in the downstream task, and the weights assigned to features during pretraining. The last one is further a function of features, noise and augmentations for pretraining data. In practical settings, we acknowledge that more factors may come into play, such as the model architecture, which varies across layers. Therefore, determining the exact optimal layer for downstream tasks is very challenging and represents an intriguing and valuable avenue for future exploration. **We believe the analytical framework established in this paper, capable of expressing downstream sample complexity in closed form through various elements in pretraining and downstream tasks, and explaining several observed phenomena (e.g., those depicted in Figure 3), can significantly aid advancing research in this direction**.

---

> ### Author Response · Authors · 2023-11-23
> **Response to Reviewer Kpzq -- Part 2**
>
> > ## “This paper demonstrates … insights about the design of the projector based on the theoretical analysis in this paper?”
>
> (1) **Layers**. The answer for the previous question has already discussed the effect of depth, where we see that the depth of the optimal layer is intricately dependent on many factors, therefore hard to decide. But the high level intuition based on that discussion is: with a greater mismatch between the pretraining and downstream tasks, more layers should be stacked in the projection head. (2) **Role of ReLu activation**. In both Theorems 3.8 and 4.2, we have shown how ReLu activation enables the pre-projection layer to learn features that are not learned at all by the post-projection layer. This occurs in both self-supervised CL, supervised CL, and standard supervised learning, whereas linear models cannot exhibit this behavior. Drawing intuition from our proof in Appendix A.4, the ReLu function exhibits a threshold-like behavior, effectively filtering the gradient passed from post-projection to prevent the elimination of the additional feature in the pre-projection layer. It could be intriguing for future research to explore whether different types of nonlinearity (e.g., GeLu, Leaky ReLu) yield varying effects and potentially facilitate a more tailored design for the projection head.
>
>
>
> > ## Validating theoretical findings on real-world datasets
>
> (1) **In terms of data augmentation's impact, empirical evidence from prior large-scale empirical studies strongly corroborates our theoretical findings**. For instance, Chen et al. (2020) Table 3, Bordes et al. (2023) Figure 2, and Rashtchian et al. (2023) Table 2 present experimental outcomes for SimCLR on large real datasets, showing that: a) information (such as Color, Hue, Rotation, and Spot) suppressed by data augmentation is more retrievable in pre-projection representations; b) consequently, if the downstream task requires such information, using pre-projection is preferable.
> (2) **For the other two scenarios**—where the downstream-relevant features are either too strong or too weak—there's been less exploration in previous research. However, **we've dedicated significant effort to provide empirical evidence**, as discussed below. It's important to note the challenge in naturally varying feature strength, as controlled modifications to images are necessary to achieve this.
> - **We've already expanded beyond purely synthetic data to include results from MNIST-on-CIFAR-10 in our original paper**. Figures 3b and 3c precisely validate our findings, including the impact of data augmentation and the non-monotonic influence of feature strength, thus taking a step closer to more realistic data representations.
> - **If the MNIST-on-CIFAR-10 dataset still seems too 'synthetic,' we conduct two new experiments on minimally modified CIFAR-10 images, which further strongly support our findings**. The experiment details can be found in Appendix D.1. In summary, we established the downstream task as categorizing CIFAR-10 images as more ‘red,’ ‘green,’ or ‘blue.’ To control the color distinguishability (the clarity in identifying a color in images), we utilized two approaches: 1) manipulating the images, or 2) selecting subsets with varying color distinguishability. This effectively controls the downstream-relevant feature’s strength. Our observations in Fig 7(b)(c) revealed that, in both scenarios, the advantage of pre-projection is more pronounced when the color feature is either too weak or too strong.
>
> “I note that the abstract on the OpenReview website is different from that on the pdf file, which should be corrected.”
>
> > ## Fixing abstract
>
> We apologize for the inconsistency, while we cannot fix it at this point, we will make sure it is reflected correctly in the final version, if accepted.

---

### Meta-Review · Area_Chair_VPS7 · 2023-12-18

**Metareview:**

Projection head has been found by many to be an effective module that enhances the feature transferability on downstream tasks, an interesting phenomenon that is widely recognized but seldomly theoretically justified. In this paper, the authors approach this problem with a (quite) simplified model and data assumptions, and show that NNs perform layer-wise reweighting and lower layer can learn more features than higher layers. They also characterize the cases when projection heads can benefit downstream tasks under their assumptions. They then extend this finding to supervised learning, showing lower layers can learn subclass-level features. Then conclude by verifying the analyses with both synthetic and real-world datasets.

Reviewers all agree that the topic of understanding projection head is of both theoretical interests and empirical significance. Meanwhile, some reviewers hold concerns on the generality of the data and model assumptions considered in this work, which might hinder its practical implications. Similarly, reviewers also raise concerns on the generality of the analysis for different projection networks, as well as for other alternative techniques. The reviewers provide thorough responses to all reviewers with theoretical and empirical evidence, which seems to address most of the concerns.

Overall, this paper contributes to a formal understanding of projection head from the feature diversity perspective, while it still has limitations in the generality of the assumptions. If accepted, the authors are encouraged to consider the reviewers’ suggestions in the final version.

**Justification For Why Not Higher Score:**

As pointed out by many reviewers, the proposed analysis has limitations in the generality of the assumption, as well as the depth of conclusions.

**Justification For Why Not Lower Score:**

Understanding the role of projection head is an important topic for representation learning and this paper contributes to a more theoretical understanding.

---

### Decision · Program_Chairs · 2024-01-16

Accept (poster)